# Global Protected Areas as refuges for amphibians and reptiles under climate change

Chunrong Mi[1,2,40], Liang Ma[3,40], Mengyuan Yang[4,5], Xinhai Li[1], Shai Meiri[6], Uri Roll[7], Oleksandra Oskyrko[1,8], Daniel Pincheira-Donoso[9], Lilly P. Harvey[10], Daniel Jablonski[11], Barbod Safaei-Mahroo[12], Hanyeh Ghaffari[13], Jiri Smid[14,15], Scott Jarvie[16], Ronnie Mwangi Kimani[17], Rafaqat Masroor[18], Seyed Mahdi Kazemi[19], Lotanna Micah Nneji[20], Arnaud Marius Tchassem Fokoua[21], Geraud C. Tasse Taboue[22], Aaron Bauer[23], Cristiano Nogueira[24], Danny Meirte[25], David G. Chapple[26], Indraneil Das[27], Lee Grismer[28], Luciano Javier Avila[29], Marco Antônio Ribeiro Júnior[30], Oliver J. S. Tallowin[31], Omar Torres-Carvajal[32], Philipp Wagner[33], Santiago R. Ron[34], Yuezhao Wang[35], Yuval Itescu[36,37], Zoltán Tamás Nagy[38], David S. Wilcove[20,39], Xuan Liu[1,41] ✉ & Weiguo Du[1,41] ✉

Protected Areas (PAs) are the cornerstone of biodiversity conservation. Here, we collated distributional data for >14,000 (~70% of) species of amphibians and reptiles (herpetofauna) to perform a global assessment of the conservation effectiveness of PAs using species distribution models. Our analyses reveal that >91% of herpetofauna species are currently distributed in PAs, and that this proportion will remain unaltered under future climate change. Indeed, loss of species' distributional ranges will be lower inside PAs than outside them. Therefore, the proportion of effectively protected species is predicted to increase. However, over 7.8% of species currently occur outside PAs, and large spatial conservation gaps remain, mainly across tropical and subtropical moist broadleaf forests, and across non-high-income countries. We also predict that more than 300 amphibian and 500 reptile species may go extinct under climate change over the course of the ongoing century. Our study highlights the importance of PAs in providing herpetofauna with refuge from climate change, and suggests ways to optimize PAs to better conserve biodiversity worldwide.

Human-induced environmental degradation is dragging global biodiversity into its sixth mass extinction[1–3]. Population and whole-species declines have rapidly spread across the animal tree of life –a phenomenon termed 'defaunation'[4–], with thousands of species on the brink of extinction and >500 species declared or believed to be extinct in the last 500 years only among terrestrial vertebrates[5–8]. Animal declines are the outcome of multiple factors operating in synergy[4,9], with anthropogenic climate change widely identified as one of the major drivers of population extirpations and whole-species extinctions in the coming century[6,10,11]. Therefore, the development of quantitative, integrative and global-scale analyses aimed at identifying the lineages (especially species) and geographic regions more likely to

undergo declines driven by climate change ranks among the major challenges for implementing effective conservation actions with the potential to mitigate these losses of biodiversity worldwide[7,12,13].

Protected Areas (PAs)—geographic regions legally designated for the protection of biodiversity and cultural resources[14]—play an essential role in maintaining global biodiversity[15,16], underpinning conservation programs worldwide to mitigate the impacts of multiple human-induced threats, including climate change[17]. Existing PAs have been designed to protect present-day biodiversity and ecosystems[18,19]. However, their effectiveness in conserving biodiversity under future climate change has only been evaluated for a few taxa, and mostly at regional scales. Therefore, the extent to which the currently established global PAs can be expected to play a dominant role under the increasing threat of climate change remains fundamentally unassessed. For instance, current models predict that PAs are likely to protect European bird populations in the face of climate change[20,21], whereas a number of areas in southern Africa are expected to become less effective for conserving endemic birds under this same threat[22,23]. Consequently, a global analysis of PA effectiveness for the conservation of species under future climate change is urgently needed to provide timely suggestions for conservation management strategies (e.g., additions to the global PA network, identification of species and regions where more intensive conservation measures such as assisted migration may be necessary, and conservation gaps more widely[24,25]).

Species distribution models (SDMs) are widely used to quantify the responses of species (e.g., rapid range shifts) under climate change[17,26]. The increasing availability of vast species occurrence datasets and environmental layers allows for the development of robust predictions of species ranges[27,28], and analyses using SDMs can clarify how their ranges have and will be affected by environmental change[17,26,29]. As expected, using species range dynamics to evaluate the role of PAs under climate change and to develop effective conservation strategies has become increasingly feasible through such analyses[18,30,31]. For example, SDMs have been used for predicting endangered species' range shifts under climate change[32–34], and for evaluating the risk of biological invasions in PAs[27,35]. SDM studies in China[18], the USA[31], Mexico[36] and South Africa[36] have revealed that current PAs may provide consistently suitable habitats under future climate conditions for tetrapods.

Amphibians and reptiles (hereafter 'herpetofauna') stand out as the most threatened terrestrial vertebrates[37,38]. In fact, amphibians are widely recognized as nature's most endangered animals[6,39]. Within both lineages, alarming proportions of species are known to be undergoing progressive population declines[2], and 41% of amphibians and 21% of reptiles are listed as facing extinction risk under IUCN red list categories[38]. In addition to land-use change[40,41], disease outbreaks[42,43], and alien species invasions[44,45], climate change has been implicated as one of the major factors involved in the decline of amphibian and reptile abundance both directly[46,47] and indirectly (e.g., by increasing susceptibility to disease[48], or enhancing demographic susceptibility to declines[49]).

The effectiveness of PAs in protecting the global herpetofauna from climate change has been evaluated in some regions and groups, such as for amphibians in China[18] and for amphibians and reptiles in the USA[31]. However, significant limitations have prevented a comprehensive global assessment of the role of PAs across these lineages worldwide. First, the availability of herpetofauna distributional data in online databases is highly biased towards high-income countries. For example, there are over 1,000,000 observed records of amphibians and reptiles in both the United States of America (USA) and Australia listed in the Global Biodiversity Information Facility (GBIF, https://www.gbif.org/), compared to only 5,327 and 22,694 observed records from mainland China and Brazil, respectively (records since 1970; accessed Aug. 15, 2022). Such huge discrepancies in data availability are a major impediment to meaningful and systematic estimations of

herpetofauna distribution and PA coverage. Second, we still lack a precise global estimation of PA coverage for herpetofauna and their predicted habitat shifts under future climate change, despite the fact that the effectiveness of PAs has been evaluated at some regional scales[18,31]. Third, most research undertaken to evaluate the effectiveness of PAs has been based on analyses using a relatively low spatial resolution (e.g., 10 km × 10 km; Supplementary Table 1). This low resolution may not be compatible with the current PA size (median = 0.37 km²), resulting in overestimations of the range size of narrow-range species. Identifying such knowledge gaps will enable us to better understand the effectiveness of PAs in conserving herpetofauna globally, and enhance our ability to protect amphibians and reptiles based on scientifically-informed decisions.

To address these limitations, we compiled a comprehensive global database with over 3.5 million filtered observation records spanning 5,403 amphibian species and 8,993 reptile species from online databases, fieldwork data, museum collections and published references. For all species in our database, we predicted the availability of suitable habitats under current (1960–1990) and future climate scenarios (2060–2080) at high spatial resolution (1 km × 1 km) using ensemble species distribution models (SDMs). Because of the limited dispersal ability of amphibians and reptiles, we assumed no occupation of newly emerged suitable habitat conditions that may become available (e.g., due to climate change) in the future[50,51]. We then evaluated the effectiveness of PAs for conserving herpetofauna by calculating species richness, range coverage of species in and out of PAs, and the proportion of species with benchmark amounts of habitats (e.g., 15% or 30% of habitat) inside PAs under present and future climate conditions (under the assumption that future land use remains unchanged for the duration of this study). Our study aims to (1) evaluate the conservation effectiveness of existing PAs in protecting herpetofauna under current and future climate scenarios, and (2) identify conservation gaps to outline a roadmap for the development of conservation actions based on the current role of PAs at a global scale. We found that the current global network of PAs already plays an important role for the conservation of current amphibian and reptile global biodiversity, and will continue to do so under future climate change. However, many species still do not occur within existing PAs and over 70% of amphibian and reptile species have under 15% of their range occurring within PAs. The conservation gaps were mainly concentrated in tropical and subtropical moist broadleaf forests, and non-high-income countries.

## Results

### Species richness inside PAs

The overall performance of our ensemble SDMs is generally good for all analyzed species (AUC = 0.95 ± 0.03, TSS = 0.87 ± 0.07; Supplementary Data 2). In terms of species richness, over 93.1% of amphibians and 91.4% of reptiles currently have suitable habitats in PAs. Additionally, 90.8% of amphibians and 90.0% of reptiles are projected to still have suitable habitats in PAs by 2070 under the RCP 4.5 scenario (Fig. 1A; Supplementary Data 3 and 4), and ≥ 90.0% under the other three RCPs scenarios (Supplementary Fig. 2). Rarity weighted richness (RWR) is predicted to increase for both amphibian and reptile species under climate change (from 89.0 to 116.0 and 147.8 to 161.9, respectively; Fig. 1B); RWR also increases under the other three RCPs scenarios, except for reptiles, which decrease under RCP 2.6 (Supplementary Fig. 2). We observed similar patterns for small-range species and IUCN threatened species (Supplementary Figs. 3 and 4), as well as for specific continents (Supplementary Figs. 5-7). When all PAs are included (Class I to VI), over 97.1% of species have suitable habitats in PAs at present, and 96.5% of all species are predicted to have suitable habitat in PAs in the future. The RWR is also predicted to increase for both amphibian and reptile species under climate change (Supplementary Fig. 8). However, our models predict that 359 to 770

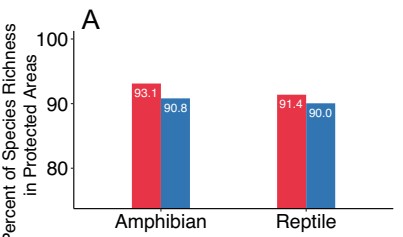
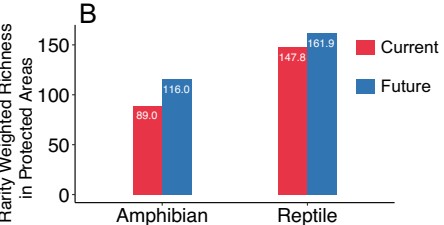

**Fig. 1 | Percent of species that have suitable habitats in protected areas (PAs).**
**A** Percent of species richness **B** the rarity weighted richness in PAs at present and by 2070 (RCP 4.5). See Supplementary Fig. 2 for the RCPs 2.6, 6.0, 8.5; Supplementary Figs. 3 and 4 for small range and threatened species; see Supplementary Figs. 5–7 for the continent region; Supplementary Fig. 8 for all species in all PAs (Class I to VI) under RCPs 2.6, 4.5, 6.0 and 8.5. We assume future land use remains unchanged for this study.

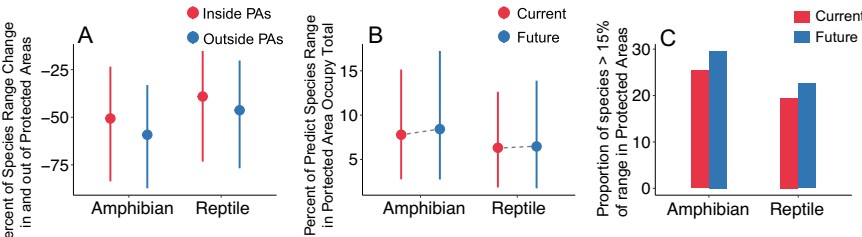

**Fig. 2 | Climate change impacts on the percentage of species range (area of habitat) inside and outside PAs by 2070 (RCP 4.5). A** Percent of species range change inside and outside PAs ($n = 5399$ and 8932 for amphibians and reptiles, respectively); **B** Percent of predicted species range in PAs at present and by 2070 ($n = 5399$ and 4876 for amphibian, 8932 and 8190 for reptile under current and future); **C** proportion of species having >15% of their range inside PAs. The points and error bars in (A) and (B) represent the medians and the 25% and 75% quantiles. See Supplementary Fig. 8 for the RCPs 2.6, 6.0, 8.5; Supplementary Figs. 9 and 10 for small range and threatened species; see Supplementary Figs. 11–13 for the continent region. We assume future land use remains unchanged for this study.

amphibian species and 545 to 1098 reptile species will go extinct under different climate change scenarios over the course of the ongoing century (Supplementary Data 3 and 4).

## Species range shifts inside and outside PAs under climate change

Under climate change, the proportion of suitable habitat projected to be lost outside PAs is significantly higher than that projected to be lost within PAs for both amphibians (median 59.2% vs. 50.6%; Wilcoxon test, $Z = 8.70$, $P \ll 0.001$) and reptiles (median 46.3% vs. 39.1%; Wilcoxon test, $Z = 8.08$, $P \ll 0.001$; Fig. 2A, under RCP 4.5 scenarios). This results in an increase in the proportion of suitable habitat inside PAs for both amphibians (from 7.8% to 8.4%; Wilcoxon test, $Z = -3.21$, $P = 0.0013$) and reptiles (from 6.3% to 6.5%; Wilcoxon test, $Z = -1.75$, $P = 0.0794$) by 2070 (Fig. 2B, under RCP 4.5 scenario). Meanwhile, the proportion of species with over 15% and 30% of their range covered by current PA networks will increase for both amphibians (from 25.4% to 29.6%, and 7.8% to 11.9%) and reptiles (from 19.4% to 22.6%, and with 6.1% to 8.2%) by 2070 (Fig. 2C; for results when the target is over 30% range in PAs see Supplementary Fig. 9). The other three RCPs scenarios present similar results (Supplementary Fig. 8). We also found similar trends for small-range species and IUCN threatened species (Supplementary Figs. 10 and 11), as well as for individual continents (Supplementary Figs. 13–15). These results indicate that PAs will provide increasingly important refuges for amphibian and reptile species under climate change.

Similar to that in strict PAs, the proportion of suitable habitat inside all PAs increase for both amphibians (from 21.0% to 22.7%; Wilcoxon test, $Z = -4.86$, $P \ll 0.001$) and reptiles (from 18.3% to 19.1%; Wilcoxon test, $Z = -2.96$, $P < 0.01$) by 2070 under climate change (Supplementary Fig. 12). The proportion of species with over 15% and 30% of their range covered by current PA networks also increases for both amphibians (from 64.4% to 66.5%, and 32.1% to 37.3%) and reptiles (from 58.4% to 59.8%, and 26.8% to 29.7%) by 2070 (Supplementary Fig. 12).

## Conservation gaps

Although our results suggest that current PAs will provide increasingly important refuges for amphibian and reptile species globally under climate change. Besides, over 1,130 species (7.9% of all species) were found to remain outside the boundaries of PAs at present or when assessed under future climate change (Supplementary Data 5). Under both current and future climate change scenarios, the amphibian family with the greatest proportion of species that are not represented in PAs is the Heleophrynidae (with 75% of species both currently and in the future not occurring in PAs); the amphibian genus with the greatest percentage of species that are not represented in PAs is *Arthroleptella* at present (100%) and *Heleophryne* in the future (100%). In reptiles, the monotypic snake family Xenotyphlopidae has the greatest percentage of species that are not represented in PAs currently (100%), while the amphisbaenian family Trogonophidae has the greatest percentage in the future (60%); at the genus level, the least protected is the Cape-Verde endemic skink genus Chioninia (100 % both currently and in the future; Supplementary Table 2).

Our analyses identify those locations with high species richness (top 20%) and small predicted species loss (bottom 20%) due to climate change by 2070 as conservation priority regions (Fig. 3A, B; 2.3% and 2.2% of global land areas for amphibians and reptiles under RCP 4.5, excluding Antarctica). Further, we identified conservation gaps as conservation priority locations not covered by existing PAs (Fig. 3C, D, 2.1% and 2.0% of land areas excluding Antarctica). These areas are mainly located in Middle America, the Tropical Andes, the north and south of South America, south and western Africa, the west coast of India, southwestern and southeastern China, Southeast Asia, and the north and southeast coast of Australia (Fig. 3C, D). In addition, large conservation gaps persist in the southeast of Canada, and the west coast and the middle of the USA for amphibians (Fig. 3C) and in the Arabian Peninsula, the eastern Mediterranean, and the northern and southwest coast of Australia for reptiles (Fig. 3D).

At the country level, we detected large conservation gap areas in the United States, Mexico, Colombia, Venezuela, Brazil, South Africa,

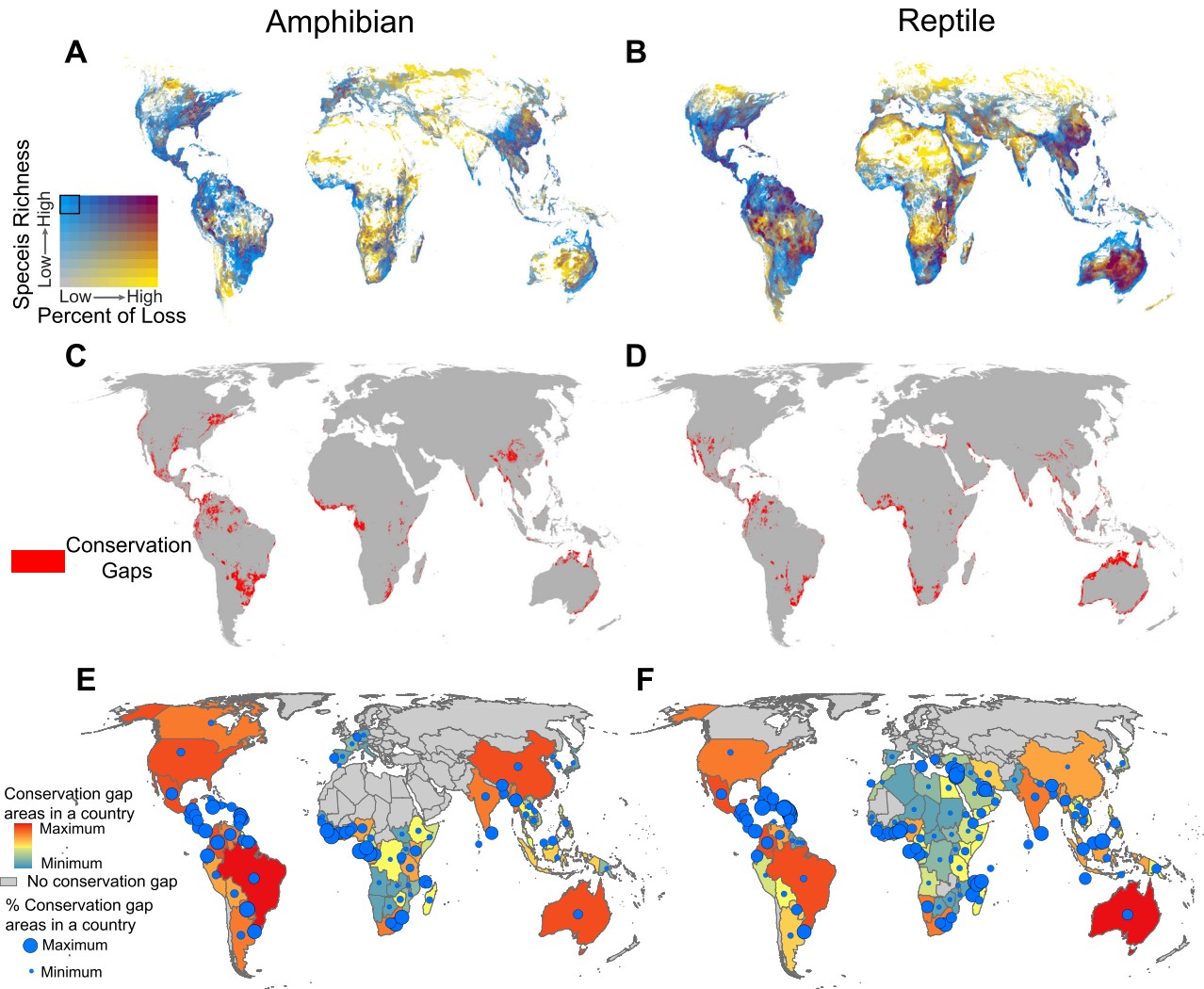

**Fig. 3 | Conservation priority and conservation gaps for global herpetofauna by 2070 (RCP 4.5).** Bivariate maps showing species richness versus percent of species loss for **A** amphibians and **B** reptiles. Each color change means a 10% quantile shift in either variable. Noteworthy, areas in blue represent areas that currently fall into the top species richness category and that are expected to suffer a low percent of species loss due to climate change by 2070; in short, they are climate-robust areas of high species richness. **C**, **D** represent conservation gaps (areas falling into the top 20% in terms of species richness and the bottom 20% in terms of future species loss due to climate change, yet fall outside the PA network of amphibians and reptiles which are outlined in black in the legend of (**A**) and (**B**). **E**, **F** conservation gaps for countries. Colors represent the area of conservation gaps in countries, and circle sizes represent the percentage of conservation gap area with respect to the land area in countries. We assume future land use remains unchanged for this study.

India, Myanmar, China and Australia for both amphibians and reptiles. Meanwhile, large conservation gaps also exist in Canada, Paraguay, Argentina, and Gabon for amphibians; and Namibia, Nigeria, and Indonesia for reptiles under RCP 4.5 (Fig. 3E, F, Supplementary Data 6). In terms of the percent of conservation gap areas per country area (i.e., the area of the sum of the gaps within a given country divided by the area of that entire country), countries in Middle America and South America (e.g., Paraguay, Panama, Uruguay, Colombia), Africa (Liberia, Gabon, Sao Tome and Principe, Eswatini and Sierra Leone) have the largest proportion of gap areas for amphibians, and countries in the Caribbean and Central and South America (e.g., Jamaica, Dominica, Uruguay), Asia (Bahrain, Israel, Siri Lanka, Lebanon) and Africa (Togo, Comoros, Benin) have the largest proportion of gap areas for reptiles (Fig. 3E, F; Supplementary Data 6). Similar results occur in the other three RCPs scenarios (Supplementary Figs. 17–19).

## Discussion

Our study provides a comprehensive, global assessment of the effectiveness of Protected Areas (PAs) in conserving reptiles and amphibians currently and in the face of future climate change over the course of the ongoing century. It is critical to emphasize that amphibians and reptiles have the highest proportions of threatened species among land vertebrate (tetrapod) groups. Understanding the effectiveness of existing PAs as a cornerstone approach for the protection of species at risk of impacts from climate change is widely regarded as one of the most important challenges to develop effective conservation strategies aimed to safeguard biodiversity[52,53], especially among highly threatened organisms. Our findings provide global-scale evidence for the crucial role that PAs play in conserving the biodiversity of amphibians and reptiles under human-induced climate change scenarios. We also filled existing gaps in the English-language species distribution data by adding information from other regions, especially South America, Africa, the Middle East, Central Asia, Russia, Pakistan and China.

Our study reveals that the proportion of amphibian and reptile habitats contained within PAs is expected to increase under climate change across the globe and in most continents (Fig. 2B, and Supplementary Figs. 9–12). We suggest that the mechanism behind this result

is that habitats outside PA boundaries will be lost due to climate change at a higher rate than habitats inside PAs (see below). Therefore, the proportion of the remaining habitats that fall within PAs will be higher after climate change than before. Furthermore, we found that the proportion of species that have benchmark amounts of habitat (e.g., 15% or 30% of habitat protected) in PAs will increase under climate change (Fig. 2C, and Supplementary Figs. 9–12). This does not mean that the proportion of habitat within PAs is sufficient to sustain demographically viable populations of amphibians and reptiles globally, nor that the areas of suitable habitat will increase in PAs, considering generally the low dispersal capabilities of these two groups. Instead, these findings suggest that larger habitat areas outside the PA network may be degraded due to the synergistic interactions of climate change with the panoply of other threats (e.g., land conversion, extractive uses of natural resources) that can be constrained within PAs, thus making current PAs even more important for amphibian and reptile species' survival in the future. Noteworthy, we projected species distribution without considering the dispersal capacity of species, and this projection may underestimate range expansions of species under climate change. The importance of PAs under climate change has also been observed across other taxa including multiple invertebrate groups and endotherms (birds and mammals)[54,55] especially when PAs are adequately managed[55]. The negative effect of climate change on bird species' distributions in Britain, Finland and the savannah region of Africa, for example, could potentially be buffered by PAs[21,54,56].

Some continental and regional scales studies have reported opposite patterns. For example, current PAs may become less effective under climate change for conserving amphibian biodiversity in Italy[57], plants in Europe[23], and birds in South Africa[22] and Southeast Asia[58]. The discrepancies among empirical studies are likely because the protection offered by PAs for species varies among taxa and regions[59]. In our study, for example, we found that PAs would become more effective for amphibians, but less effective for reptiles in South America under RCP 2.6 and 4.5 (proportion of species with 30% of habitat protected), and more effective for both amphibians and reptiles in Africa (Supplementary Fig. 16). In addition, this discrepancy might also be caused by different conservation targets and RCP scenarios. For example, European PAs are predicted to become more effective for reptiles under all four RCPs scenarios when the target is 15% of species habitat within PAs, whereas Europeans PAs would be less effective under RCPs 2.6 and 4.5 when the target moves to 30% (Supplementary Fig. 16). Overall, most regional studies are consistent with our results showing that PAs will be critically important to protect herpetofauna under climate change globally, despite variation in geographical focus, taxonomic groups explored, PA types, conservation target scopes, and climate scenarios variations.

One of the potential explanations for the increasing effectiveness of PAs is that the climate is less extreme inside PAs than outside (Supplementary Fig. 20). For example, local extinctions are expected to occur faster in extremely hot temperature regions[13], and most PAs are associated with a lower rate of climate change[53,60] which may mitigate climate change impacts for communities inside PAs[60]. Additionally, precipitation in PAs is higher than outside and is predicted to increase in the future (Supplementary Fig. 20). Previous studies have found that precipitation is highly correlated with reptile and, especially amphibian distribution and abundance[61–63]. This, together with the relatively mild climate inside PAs may mitigate the negative impacts of climate change on species[60,64,65]. However, over 300 amphibian and 500 reptile species are predicted to go extinct due to climate change over the course of the ongoing century. These were not counted when we calculate the proportion of species covered in PAs in the future, hence our finding—that a large majority of species will be protected in the future relates to surviving species and should not be taken to mean that climate change will not have devastating effects on many amphibian and reptile species.

Our evidence shows that the current global network of PAs already plays an important role for the conservation of current amphibian and reptile global biodiversity, and will continue to do so under predicted future climate. However, many species do not occur within existing PAs. These include many amphibians and reptile species distributed in Mexico, Jamaica, the Andes Mountains, western Africa, South Africa, the south and north coast of Turkey, Yemen, western Iran, and the eastern Papua New Guinea (Supplementary Figs. 21 and 22). Additionally, more conservation gaps are identified in southern USA, Ethiopia, northwestern Congo for amphibians, and Argentina, Somalia, India, Australia for reptiles (Supplementary Fig. 22). Furthermore, we did not analyze 36.4% and 23.6% of global amphibian and reptile species given the limited availability of robust distributional data (many of which are likely to have ranges entirely outside of PAs). In general, unprotected species have small geographic range and urgently need to be covered by PAs (Supplementary Fig. 23), given that small-ranged species are disproportionately threatened[6,66,67], and are under high extinction risks due to climate change[68].

In addition, we found that over 70% and 77% of amphibian and reptile species, respectively, have under 15% of their range occurring within the boundaries of strict PAs at present and in the future (Fig. 2C). This percentage increase to 88.1% and 91.8% of amphibian and reptile species, respectively, when we use a threshold of at least 30% of the range in PAs (Supplementary Fig. 9). Thus, a strategic expansion of PA networks for these organisms is urgently needed.

The areas of land in PA status increases 1.2 times when we expand our focus from current strict PAs (Class I to IV) to encompass all PAs (Class I to VI). As a result, the proportion of species with >15% or >30% of their habitat in PAs would increases by 1.2–2.0 times and by 2.1–3.4 times at present and under future climate conditions, respectively (Supplementary Figs. 9 and 12). This suggests that the management of less strictly protected PAs (Class V and VI) is of paramount importance for increasing the protection effectiveness of PAs in protecting the herpetofauna globally. Indeed, previous studies have shown that PAs management contributes significantly to conservation. For instance, active management of PAs enhances metapopulation expansion of *Hesperia comma* under climate change[55], and PAs managed for global waterbirds and their habitat are more likely to benefit populations than are unmanaged PAs[69].

Our study found that conservation gaps were mainly concentrated in tropical and subtropical moist broadleaf forests (45.9% and 31.6% of range for amphibians and reptiles under RCP 4.5; Supplementary Fig. 24). Such forests are characterized by low variability in annual temperature, high levels of rainfall and rich tree communities, which all contribute to the high levels of amphibian and reptile diversity found in those areas[70]. The expansion of the global PAs network in these regions is necessary to protect their rich and unique herpetofauna. The unprotected regions (Fig. 3C, D), i.e., the areas we identified as key conservation gaps, have also been identified as being globally significant for other taxa[67,71]. These regions should be the priority locations for future PA adjustments to better conserve biodiversity[72].

The wide distribution of amphibians and reptiles may lead to some regions being ignored. Thus, analyses of conservation gaps at the country scale will contribute to better biodiversity protection and management, along with a more efficient allocation of conservation resources[67]. We found large conservation gaps for amphibians and reptiles in countries such as Brazil, Mexico, Colombia, South Africa, India, Myanmar and China (Figs. 3E and 3F, Supplementary Data 6). These are not high-income countries, which may affect the amounts of resources available for conservation actions (Supplementary Fig. 25). This is a classic conflict between biodiversity conservation and

economic development[73,74]. Surprisingly, some high-income countries also have large gaps, such as Canada for amphibians, and the USA and Australia for both amphibians and reptiles. These three countries are large, and expanding PAs to encompass more conservation areas should be easier to accomplish than in those countries with small land areas or low income. Our maps may help countries fulfill their Post-2020 Global Biodiversity Framework[75].

Global amphibian and reptile biodiversity is projected to decline significantly under ongoing climate change. Given these alarming expectations, our finding that most amphibian and reptile species that will survive climate change have at least part of their ranges located within the current PA network offer promising scenarios that reinforce the effective effects that legislation has on the mitigation of extinctions worldwide. We also predicted that the percentage of species' ranges inside PAs will increase under climate change. This is largely a function of higher predicted rates of habitat loss due to climate change outside PAs versus inside PAs. However, our models still predict that hundreds of species from both classes of ectotherms will be lost to climate change both in reserves and, especially, outside of them. Moreover, many species are only found in reserves with low degrees of protection from anthropogenic disturbance (i.e., type V and VI reserves). Thus, our findings should not be interpreted as a cause for complacency. We identified current PAs which need expansion to cover vulnerable species that may be particularly sensitive to climate change. Current PAs already provide important refuges for the conservation of herpetofauna under future climate change scenarios, but their conservation effectiveness could be further enhanced by better protection in less-strict PAs and the establishment of (strict) reserves in the conservation gap areas we have identified. In addition, more occurrence records collected for data-deficiency species in future and the update of PAs from the WDPA database may influence species distributes in PAs and therefore optimal conservation plans. Finally, it is necessary to highlight that one potential caveat with the present study is that many rare, small-ranged, species were excluded from our analyses as they currently lack sufficient distributional records to construct SDMs. Most of these unanalyzed species, 70.5% of amphibians and 64.7% of reptiles, are assessed as threatened or data deficient according to IUCN (compared to only 22.9% of amphibians and 16.3% of reptiles included in our dataset). Consequently, further attention to the plight of these species is thus needed when the importance of PAs for their conservation is assessed in the future, because these species are more likely to be at a high extinction risk.

## Methods
### Occurrence records
We searched for occurrence records of amphibians and reptiles from online databases, fieldwork, museum records and data published in the primary scientific literature (e.g., see protocols in Roll et al.[76], reference source see Supplementary Data 1). First, we collected all occurrence records from 1970–2022 with precise geographic information (i.e., longitude and latitude) from GBIF (https://www.gbif.org/; accessed in May 2022), iDigBio (https://www.idigbio.org/), and VertNet (http://vertnet.org/). Next, we conducted intensive data collection to supplement global occurrence records, especially in those underrepresented regions in the traditional databases and literatures, such as Central Asia to eastern Europe (e.g., Uzbekistan, Kazakhstan, Kyrgyzstan, Turkmenistan, Russia, Ukraine, Latvia, Hungary), South Asia (Pakistan, India), the Middle East (Turkey, Iran, the Arabian Peninsula), Africa (Chad, North and South Sudan, the republics of the Congo, Tanzania, Kenya, Nigeria, Cameroon), and South America (Bolivia, Paraguay, Argentina, Peru, Colombia). Although it might inflate the area of species distribution, we removed records from the occurrence records that fell outside the 400 km buffer of the species polygon maps following Ficetola et al[77]. to correct potential errors of occurrence records in database (amphibians distribution maps prepared by

the International Union for Conservation of Nature and Natural Resources (IUCN) accessed in March 2019; reptile distribution maps referred the latest available reptile distribution map[76] using R 3.6.2. We then combined all datasets and used the 'CoordinateCleaner' package implemented in R to remove records from capitals, institutes and museums[49]. Next, we used 'spThin' package in R[78] to minimize sampling bias by filtering occurrences within a single grid cell (1 km × 1 km)[79]. Species with ≥5 presence records were selected for further analyses[18,80]. Our cleaned dataset included data for 5403 amphibian species with a total of 1,386,788 occurrence records and 8993 reptile species with a total of 2,163,074 occurrence records (Supplementary Fig. 1). Our dataset covers 63.6% (5403/8,489) of amphibians and 76.6% (8993/11,733) of reptiles known to date. However, many rare, small-ranged, species were not included in this study as they do not currently have enough occurrence records available. These species are already more threatened than included species (see above), are probably less well covered by existing PAs, and may be more vulnerable to range shifts from climate change[81]. Our results are thus mostly applicable for the 64–77% of species with overall larger ranges.

### Environmental predictors
We focused on climate change scenarios, but do not address land-use change. We extracted climatic variables representing current (1960–1990) and future (2060–2080) climate conditions at the 30 s (~1 km × 1 km) resolution from the WorldClim database[82]. In total, we used six bioclimatic variables to construct our species distribution models: annual mean temperature (BIO1), Isothermality (BIO3; Mean Diurnal Range/Temperature Annual Range), maximum temperature of the warmest month (BIO5), minimum temperature of the coldest month (BIO6), mean annual precipitation (BIO12), and precipitation during the warmest quarter (BIO18). These variables were selected given that they represent the climatic components that constrain the physiology, reproduction and life-history traits of amphibian and reptile species and are, therefore, key determinants of their distribution[61,83]. To test for multicollinearity in our SDMs, we used the 'vif' function in the 'usdm' R package to remove variables with a Variance Inflation Factors <10[84] from the initial six bioclimatic variables for each species.

To account for uncertainty in projections of future climates, we considered three Global Circulation Models (GCMs): BCC-CSM1-1 developed by Beijing Climate Center[85], CNRM-CM5 developed by National Centre for Meteorological Research and Centre Européen de Recherche et de Formation Avancée en Calcul Scientifique[86], and MIROC-ESM developed by the Japan Agency for Marine-Earth Science and Technology[87]. We used four Representative Concentration Pathways (RCPs) 2.6, 4.5, 6.0 and 8.5 as future climate conditions during 2060–2080 (2070)[83]. We selected these four scenarios because they span a wide range of plausible global change futures, and serve as the basis for climate model projections[28,51]. We averaged the climate data across the three GCMs for each grid to reduce uncertainties among modeling techniques[29,59,88]. We projected all climate layers to Eckert IV equal-area projection[67] with 1 km × 1 km grid resolution.

### Species distribution models
We used the 'sdm' package[89] to implement an ensemble species distribution model to predict species distributions in R with a high-performance cluster. For our input data, we generated pseudo-absence records using the 'gRandom' method[89], within a calibration area using Wallace's Zoogeographic Regions[90] where species occurrence records (presence points) are located and delimited by a buffer of 500 km around the presence records[91–93]. We then used a 70% random sample of initial data (presence-absence) as training data and evaluated them against the remaining 30% of samples. We repeated split sampling five times to account for the uncertainty associated with

data partition[94]. In sum, we used five commonly used and with high model performance SDM algorithms in the ensemble models: Generalized Linear Model[95], Generalized Boosted Regression Models[96], Maximum Entropy[97], Random Forest[98] and Support Vector Machines[99].

To evaluate the performance of our models, we used true skill statistics (TSS)[100] and the area under the receiver operating characteristic curve (AUC)[101]. We only retained the model when TSS ≥ 0.7[102,103], and then created an ensemble model that weighted single models by their model performance (TSS) for each species[102–104]. Species habitat suitability maps were transformed to binary distribution maps (presence/absence) with the threshold that maximizes TSS, which has been widely used in producing species potential distribution maps[29,91]. We made a model flow to describe how we run SDM model (see Supplementary Example).

### Statistical analyses
We used PAs from the July 2022 World Database of Protected Areas (WDPA) dataset[105], a resource widely used in conservation biology and biogeography studies related to PAs on a global scale[106,107]. Because all national PAs for China are not included in the WDPA dataset after 2018[107,108], we combined the 2022 WDPA layer with PAs in China from the April 2018 WDPA[106,107]. The polygons in the WDPA database have been classified into six types with conservation functions: strict nature reserves (Ia), wilderness areas (Ib), national parks (II), natural monument or feature (III), habitat/species management area (IV), protected landscape/seascape (V), protected area with sustainable use of natural resources (VI), and those not assigned permitting certain human activities. We compared the impact of all PAs with that of strict PAs[109] (Class I to IV) in the subsequent analyses. We overlaid SDM predictions (1 km × 1 km) with PA maps. We then quantified the predicted species richness inside PAs, and calculated the proportion of species inside any PAs under current and future climate scenarios. We also calculated species richness by adopting a rarity weighted richness measure (RWR);[110–112] this method highlights weighting of species with small ranges. The RWR inside PAs was calculated in two steps: (1) each species was given a score (calculated as the inverse of the range size where it occurs), and (2) the RWR inside PAs is the sum of individual scores of all species occurring in PAs[113]. Next, we calculated the area of suitable species habitat (range where the species is predicted to occur by the SDM models) inside and outside PAs under current and future climate scenarios. Also, we calculated the proportion of species ranges inside and outside PAs under current and future climate conditions. Additionally, we reported the proportion of species with >15% of their range covered by any PAs[31,114]. Here we treat the 15% coverage as a summary benchmark of conservation status (i.e., 15% of a species' predicted suitable habitat must be protected); further, we changed this target to 30% by way of a sensitivity test. These analyses were conducted for (1) all species (5403 amphibians and 8993 reptiles), (2) small-range species (i.e., with distributional ranges smaller than the median value of all species; 2697 and 4,496 species for amphibians and reptiles, respectively), and (3) threatened species (i.e., classified as Near Threatened, Vulnerable, Endangered, Critically Endangered, and Extinct based on IUCN; 1,277 and 1,490 species of amphibians and reptiles, respectively). We conducted analyses at both global and continental scales. Finally, we identified areas with the top 20% species richness and bottom 20% of predicted species loss[115,116] as conservation priority areas with 1 km × 1 km resolution and overlaid these areas with current PAs. This allowed us to identify the portion of those conservation priority areas that fell outside the boundaries of PAs as conservation gaps, providing valuable information to guide efforts for expanding PA networks in the future. All our analyses were at a spatial grain of 1 km × 1 km, ≥100 times finer than used in most previous regional assessments (Supplementary Table 1). This represents a significant improvement as it brings the scale of our analysis closer to the sizes of PAs (median = 0.37 km²), therefore providing realistic guidance for

conservation. Most amphibian and reptile species, especially salamanders, have weak dispersal abilities[50,51], the dispersal distance differs among populations[50], which are difficult to be controlled in SDM construction. We therefore calculated all metrics assuming no dispersal capacity under four RCP scenarios (RCPs 2.6, 4.5, 6.0, and 8.5). This means that, under our assumptions, no range expansions can occur. We report the results under RCP 4.5 in the main text and provide the results under the other three RCP scenarios in the Supplementary Information. All statistical tests were conducted using the Wilcoxon test in R 3.6.2.

### Data availability
All online occurrence records are available at https://doi.org/10.6084/m9.figshare.20958190.v1. Some occurrence records are available under restricted access for avoiding potential threat of poaching, access can be obtained by contacting the data owners, who have been listed in our Supplementary Data 7. Climate change impact data, protected area coverage data generated in this study are provided as Supplementary Data 3 and 4. Climate data are from WorldClim database (www.worldclim.org). Maps of the spatial distribution of amphibian species from IUCN: www.iucnredlist.org/resources/spatial-data-download, reptile species from Roll et al.[76] Distribution of protected areas are from World Database on Protected Areas (WDPA, www.protectedplanet.net, accessed July 2022), PAs of China from the April 2018 WDPA. Wallace's Zoogeographic Regions are from Holt et al.[90].

### Code availability
The code used in the analyses is available at the following public repository: https://doi.org/10.6084/m9.figshare.20958190.v1.

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

## Acknowledgements

We thank all the collaborators and their teams for sharing their precious
data so that our research can be completed. We thank the comments
from Baojun Sun. This research is funded by National Natural Science
Foundation of China (No. 31720103904, 31821001, 31870391). X.L. was
supported by the Third Xinjiang Scientific Expedition Program
(2022xjkk0800 and 2021xjkk0600), and the grants from Youth Innova-
tion Promotion Association of Chinese Academy of Sciences (Y201920).
D.P.D. was supported by a grant provided by the School of Biological
Sciences at Queen's University Belfast. D.J. was supported by the Slovak
Research and Development Agency under the contract APVV-19-0076
and by the grant VEGA 1/0242/21 of the Scientific Grant Agency of the
Slovak Republic. J.S. was supported by the Charles University Research
Centre program No. 204069. L.M.N. received funding for field surveys in
Nigeria from Rufford Foundation (Grant Nos: 22507-1, 29951-2) and
National Geographic Early Career Grant (EC-357C-18).

## Author contributions

C.M., L.M., X.L. and W.D. wrote the manuscript. C.M. and L.M. designed
and performed all data analyses, M.Y. performed the parallel comput-
ing. X.L. W.D, S.M., U.R. and D.P.D. edited the manuscript. J.S. and S.J.
run models for the species in the Arabian Peninsula and New Zealand,
respectively. C.M., X.L, S.M., U.R., O.O., D.P.-D., L.P.H., D.J., B.S.-M., H.G.,
J.S., S.J., R.M.K., R.M., S.M.K., L.M.N., A.M.T.F., G.C.T.T., A.B., C.N., D.M.,
D.G.C., I.D., L.G., L.J.A., M.A.R.-J., O.J.S.T., O.T.-C., P.W., S.R.R., Y.W., Y.I.,
Z.T.N. collected occurrence data. W.D. and X.L. supervised the study.

## Competing interests

The authors declare no competing interests.

### Ethics

Our surveys were reviewed and approved by the Animal Ethics Com-
mittee at the Institute of Zoology, Chinese Academy of Sciences
(IOZ14001).

## Additional information

**Supplementary information** The online version contains
supplementary material available at

Xuan Liu or Weiguo Du.

**Peer review information** *Nature Communications* thanks David Bickford,
Amael Borzee, Babak Naimi, Tongli Wang and the other, anonymous,
reviewer for their contribution to the peer review of this work. Peer
reviewer reports are available.

[1]Key Laboratory of Animal Ecology and Conservation Biology, Institute of Zoology, Chinese Academy of Sciences, Beijing, China. [2]University of Chinese
Academy of Sciences, Beijing, China. [3]School of Ecology, Shenzhen Campus of Sun Yat-sen University, Shenzhen, China. [4]Zhejiang University,
Hangzhou, China. [5]Westlake University, Hangzhou, China. [6]School of Zoology and Steinhardt Museum of Natural History, Tel Aviv University, Tel Aviv, Israel.

[7]Mitrani Department of Desert Ecology, The Jacob Blaustein Institutes for Desert Research, Ben-Gurion University of the Negev, Midreshet Ben- Gurion, Israel. [8]Educational and Scientific Center, Institute of Biology and Medicine, Taras Shevchenko national University of Kyiv, Kyiv, Ukraine. [9]School of Biological Sciences, Queen's University Belfast, Belfast, UK. [10]School of Science and Technology, Nottingham Trent University, Clifton Campus, Nottingham, UK. [11]Department of Zoology, Comenius University in Bratislava, Bratislava, Slovakia. [12]Pars Herpetologists Institute, Corner of third Jahad alley, Arash Str., Jalal-e Ale-Ahmad Boulevard, Tehran, Iran. [13]Department of Environmental Sciences, Faculty of Natural Resources, University of Kurdistan, Sanandaj, Iran. [14]Department of Zoology, Faculty of Science, Charles University, Prague, Czech Republic. [15]Department of Zoology, National Museum in Prague, Prague, Czech Republic. [16]Otago Regional Council, Dunedin, 9016 Aotearoa, New Zealand. [17]National Museum of Kenya, Herpetology Section, Nairobi, Kenya. [18]Zoological Sciences Division, Pakistan Museum of Natural History, Garden Avenue, Shakarparian, Islamabad, Pakistan. [19]Zagros Herpetological Institute, Somayyeh Avenue, Qom, Iran. [20]Department of Ecology and Evolutionary Biology, Princeton University, Princeton, NJ, USA. [21]Laboratory of Zoology, University of Yaoundé, Yaoundé, Cameroon. [22]Multipurpose Research Station, Institute of Agricultural Research for development, Bangangté, Cameroon. [23]Department of Biology and Center for Biodiversity and Ecosystem Stewardship, Villanova University, Villanova, PA, USA. [24]Departamento de Ecologia, Instituto de Biociências, Universidade de São Paulo, São Paulo, Brazil. [25]Royal Museum for Central Africa, Tervuren, Belgium. [26]School of Biological Sciences, Monash University, Clayton, VIC, Australia. [27]Institute of Biodiversity and Environmental Conservation, Universiti Malaysia Sarawak, Sarawak, Malaysia. [28]Department of Biology, La Sierra University, Riverside, CA, USA. [29]Grupo Herpetología Patagónica (GHP-LASIBIBE), Instituto Patagónico para el Estudio de los Ecosistemas Continentales (IPEEC-CONICET), Puerto Madryn, Argentina. [30]Department of Zoology, Tel-Aviv University, Tel-Aviv, Israel. [31]UN Environment Programme World Conservation Monitoring Centre, Cambridge, UK. [32]Museo de Zoología, Escuela de Ciencias Biológicas, Pontificia Universidad Católica del Ecuador, Quito, Ecuador. [33]Allwetterzoo, Münster, Germany. [34]Museo de Zoología, Escuela de Biología, Facultad de Ciencias Exactas y Naturales, Pontificia, Universidad Católica del Ecuador, Quito, Ecuador. [35]Chengdu Institute of Biology, Chinese Academy of Sciences, Chengdu, China. [36]Leibniz Institute of Freshwater Ecology and Inland Fisheries (IGB), Müggelseedamm, Berlin, Germany. [37]Institute of Biology, Freie Universität Berlin, Berlin, Germany. [38]Independent researcher, Berlin, Germany. [39]Princeton School of Public and International Affairs, Princeton University, Princeton, USA. [40]These authors contributed equally: Chunrong Mi, Liang Ma. [41]These authors jointly supervised this work: Xuan Liu, Weiguo Du. ✉e-mail: liuxuan@ioz.ac.cn; duweiguo@ioz.ac.cn

