## [Peer Review File · Nature Communications]

Reviewer comments, first round review –

Reviewer #1 (Remarks to the Author):

This manuscript presents the first study that examined the effectiveness of PAs in conserving herpetofaunal richness and ranges under current and projected climate conditions on a global scale at a fine spatial resolution. It also identified conservation gaps, which is important for effectively expanding the PA network. The issues addressed by this study are broadly concerned and timely important. The data used in this study were comprehensive, the analyses were solid, the results were clearly presented, and the conclusions were well supported by the results. The manuscript was well-organized, concisely written, and easy to follow. I really enjoyed reading it, and it is one of the best papers that I have read this year. I strongly recommend it be published in Nature Communications after the following minor issues are fixed.

Minor issues:

Lines 105-106: it is unclear about the grid size and the number of occurrences kept after the filtering.

Lines 127-128: Even though the statistical trends were similar, the magnitude of the loss of habitats due to climate change can be very different between RCP 2.6 and RCP 8.5. Thus, the loss of habitats under RCP 8.5 should also be included in the results and discussion.

Line 170: Why to use 10 km x 10 km resolution here instead of 1 km x 1 km?

Line 350: potential PAs?

Reviewer #2 (Remarks to the Author):

[Editor's note: see also the attached annotated document]

I would like to start by praising the authors for the work conducted here, this is a major achievement, and something much needed from a conservation research perspective. I have some points that I think are critical to address. Please see below, but also see the file attached as I have principally commented on the word file. I am also signing this review, and I am available for further revisions if the journal considers it relevant.

Major comments

1) There is a large problem regarding the selection of species, and the conclusions drawn from the dataset. This is half of the species for both amphibians and reptiles! And the most abundant species are the one for which more datapoints are available, and thus the ones selected here, so it totally biases the data. You conclude on a high number of species in PAs under climate change scenario, but this mostly includes widespread species (or well known, and therefore more likely to already be in PAs). As a result, you cannot speak about global herpetofauna in this case, this study is about the abundant herpetofauna, and the title, abstract, and discussion are currently misleading, and potentially extremely detrimental to conservation.

2) BioClim 3 (isothermality) is critical for many ectotherms. Why wasn't the variable considered here?

3) We know that RCP 2.6 has already been exceeded, and RCP 4.5 and 6.0 are the most likely scenario, but these scenarios have been ignored, and the author went for RCP 8.5 instead (quite an extreme – and the results are not presented). This is not representative of species' ecology, and it is now established that there are variations between RCPs (you can easily find a few examples following my name as I am signing the review, but I don't want to provide a link as not to bias the author's choice in their citations of the literature).

4) The authors overlaid SDM predictions (1 km × 1 km) with PA maps downloaded from the World Database of Protected Areas (WDPA, <http://www.protectedplanet.net>, accessed July 2017), and there is a problem here. Not all areas from the database are PA strictly speaking. Some are areas with regulations, but they are not PAs.

5) One of the models include "species with unlimited dispersal capacity". I only want to ask how

biologically relevant is that. Some salamanders can disperse a few metres at most. There are averages dispersion value available, please use them.
Based on these biases, it is hard to comment on the discussion or specific results.

Minor comments:

- 1) Why making such a big deal about the addition of Chinese species? There shouldn't be a political agenda in scientific research, and I would recommend the authors to do the same for other regions where the GBIF dataset is under-represented if they want to maintain their claims.
- 2) Authors state that "For records with only location information (village or town), we georeferenced the longitude and latitude using Google Earth v7.1". But, If you use a 1 x1 km km2 scale, then this does not work to build models. Your species will be expected to thrive in urban habitat (and the correction for that later in the text only includes "large cities". We cannot judge on the quality of the data.
- 3) You cannot compare families by the number of species in PA, take the Scincidae for instance, that's about 4000 species (more or less), how can it be compared to family that have a few dozen (or even hundreds) of species. Please use ratios instead.
- 4) Some minor mistakes in English, related to typos or missing words more than structure.

Amael Borzee

Reviewer #3 (Remarks to the Author):

The manuscript focused on an important topic about evaluating the effectiveness of protected areas (PAs) for amphibians and reptiles on a global scale. In general, the study is interesting as it involved nice efforts to collect species records from multiple resources/studies and followed a sophisticated workflow to quantify the effectiveness of PAs and identify gaps in the baseline climate condition and under future climate scenarios. The results, however, seem surprising to me as I found big inconsistencies between them and the results of a similar study (but at the regional level) that we conducted in our lab (not published yet). The authors need to explore and discuss the consistencies/inconsistencies of their results and the other studies (which are missing from the manuscript).

One methodological issue that may affect the results is that the SDMs are strongly sensitive to extent of the study area where background (pseudo-absences) records are drawn. To address the issue, the extent can be limited to the areas that are accessible for species to draw background points for each species separately. It means that for each species, the SDMs would be trained using records within the accessible areas. One way to approximate those areas for each species is to use Wallace zoogeographic regions and consider areas where species occurrence records (presence points) are located. The trained SDMs over a limited extent can then be used to predict/project the potential distribution on a global scale as far as the areas have a climate condition within the range used to train the model (i.e., the areas with a condition outside of the range should be excluded from the predicted map as the SDMs do not perform well for extrapolations).

Some minor issues:

Lines 126-127: What do you mean by "the statistical trends of RCP 2.6 and 8.5 were similar"? Do you mean that the effectiveness of PAs would not be affected under the worst future climate scenario compared to "very stringent" pathway (RCP 2.6)?

Line 133: the "eRandom" method is not activated in the sdm package yet, so it means that you used the "gRandom" method (default).

Line 144: modify the sentence "... the threshold by maximum TSS" to "...the threshold that maximizes TSS".

It would be nice if the codes used to run the workflow are shared in the supplementary, which could be used for a more precise evaluation of the workflow.

**Global Protected Areas as refuges for amphibians and reptiles under climate change
(NCOMMS-22-13511-T)**

Comments from reviewers 1:

This manuscript presents the first study that examined the effectiveness of PAs in conserving herpetofaunal richness and ranges under current and projected climate conditions on a global scale at a fine spatial resolution. It also identified conservation gaps, which is important for effectively expanding the PA network. The issues addressed by this study are broadly concerned and timely important. The data used in this study were comprehensive, the analyses were solid, the results were clearly presented, and the conclusions were well supported by the results. The manuscript was well-organized, concisely written, and easy to follow. I really enjoyed reading it, and it is one of the best papers that I have read this year. I strongly recommend it be published in Nature Communications after the following minor issues are fixed.

Response: Thank you very much for your positive and constructive comments. We have addressed all your concerns one by one below.

Minor issues:

1. Lines 105-106: it is unclear about the grid size and the number of occurrences kept after the filtering.

Response: Thanks for pointing this out. We have added this information in our revised manuscript “Next, we used ‘spThin’ package in R¹ to minimize sampling bias by filtering occurrences within a single grid cell (1 km × 1 km)². Species with ≥5 presence records were selected for further analyses^{3,4}”, **Line 379-381**.

2. Lines 127-128: Even though the statistical trends were similar, the magnitude of the loss of habitats due to climate change can be very different between RCP 2.6 and RCP 8.5. Thus, the loss of habitats under RCP 8.5 should also be included in the results and discussion.

Response: Thanks for the suggestion, we have re-run SDMs under scenarios of RCP 2.6, 4.5, 6.0 and 8.5. As you will see, we obtained similar patterns under different scenarios, although there were some differences among RCPs, which is consistent with your suggestions. We have described and addressed this issue in the **Methods, Results and Discussion** sections. Please see more details in **Line 174, 177, 192-193, 237, 276, 279, 403, 464-466**.

3. Line 170: Why to use 10 km x 10 km resolution here instead of 1 km x 1 km?

Response: We used 10 km × 10 km resolution here instead of 1 km × 1 km, simply because we did not have access to a high-performance computer with enough running memory. We have now solved this problem by using a more powerful computer cluster with 1.5T RAM. Now all conservation gap analysis is based on 1 km x 1 km resolution.

4. Line 350: potential PAs?

Response: Thanks for the suggestion, we have changed it to “current PAs”, **Line 351**.

Comments from reviewers 2:

I would like to start by praising the authors for the work conducted here, this is a major achievement, and something much needed from a conservation research perspective. I have some points that I think are critical to address. Please see below, but also see the file attached as I have principally commented on the word file. I am also signing this review, and I am available for further revisions if the journal considers it relevant.

Response: We appreciate very much the reviewer's positive and constructive comments. We have addressed all your concerns and comments below.

Major comments

1. There is a large problem regarding the selection of species, and the conclusions drawn from the dataset. This is half of the species for both amphibians and reptiles! And the most abundant species are the one for which more datapoints are available, and thus the ones selected here, so it totally biases the data. You conclude on a high number of species in PAs under climate change scenario, but this mostly includes widespread species (or well known, and therefore more likely to already be in Pas). As a result, you cannot speak about global herpetofauna in this case, this study is about the abundant herpetofauna, and the title, abstract, and discussion are currently misleading, and potentially extremely detrimental to conservation.

Response: To solve the problem of potential data bias, we have tried our best to invite over 30 international herpetologists as co-authors and expanded the dataset by collecting occurrence records of an additional 1048 amphibian and 4136 reptile species across the globe. Our current dataset includes 5403 amphibian and 8993 reptile species, which has covered 63.6% and 76.6% of all amphibian and reptile species that have been recorded so far. To our knowledge, this is the most comprehensive dataset with available and precise amphibian and reptile occurrence data at a fine spatial resolution for SDM constructions now.

We have included those species with ≥ 5 occurrence records in our analysis, and thereby our analysis not only included widespread (abundant) species, but also many small-range (rare) species. More importantly, our conclusions did not change even after we expanded our dataset, demonstrating that our dataset is not biased and our conclusions are independent of sampled species and reliable. Nonetheless, we agree with the reviewer's opinion that some extremely rare species are excluded from our analysis due to their occurrence records are currently

unavailable. We have added sentences in the **Discussion** section (**Line 300-302, 356-359**) to further address this issue.

2. BioClim 3 (isothermality) is critical for many ectotherms. Why wasn't the variable considered here?

Response: Following the reviewer's very good suggestion, we have now included isothermality (BIO3) as a predictor and re-run all our SDMs. Importantly, we obtained similar results after incorporating this variable.

3. We know that RCP 2.6 has already been exceeded, and RCP 4.5 and 6.0 are the most likely scenario, but these scenario have been ignored, and the author went for RCP 8.5 instead (quite an extreme – and the results are not presented). This is not representative of species 'ecology, and it is now established that there are variations between RCPs (you can easily find a few examples following my name as I am signing the review, but I don't want to provide a link as not to bias the author's choice in their citations of the literature).

Response: Thank you for pointing out this key question. We have re-run our SDMs under all four RCP scenarios, and state in the **Methods, Results and Discussion** sections (**Line 174, 177, 192-193, 237, 276, 279, 403, 464-466**). “We used four Representative Concentration Pathways (RCPs) 2.6, 4.5, 6.0 and 8.5 as future climate conditions during 2060–2080 (2070)⁵. We selected these four scenarios because they span a wide range of plausible global change futures, and serve as the basis for climate model projections^{6,7}” (**Line 402-405**). As you will see in our revised manuscript, although we obtained an overall similar pattern, more species range will be predicted loss both inside and outside PAs from RCP 2.6 to 8.5. For the rarity weighted richness, the proportion of suitable habitat in PAs and the percentage of effectively protected species will both be predicted to increase under all four RCPs, which indicate that PAs will be more important as refugia of amphibians and reptiles under climate change.

4) The authors overlaid SDM predictions (1 km × 1 km) with PA maps downloaded from the World Database of Protected Areas (WDPA, <http://www.protectedplanet.net>, accessed July 2017), and there is a problem here. Not all areas from the database are PA strictly speaking. Some are area with regulations, but they are not PAs.

Response: Thank you for this input. We used strict PAs⁸ (Class I to IV) to re-analyze our data. Additionally, we compared the impact of all PAs (Class I to VI) with that of strict PAs

in the subsequent analyses (**Line 426-435**). We found a similar pattern under climate change (**Line 179-181, 197-202**).

5) One of the models include “species with unlimited dispersal capacity”. I only want to ask how biologically relevant is that. Some salamanders can disperse a few metres at most. There are averages dispersion value available, please use them. Based on these biases, it is hard to comment on the discussion or specific results.

Response: Thank you for pointing out this important issue. We agree that most amphibians and reptiles have relatively limited natural dispersal abilities. In fact, there will be geographical variations in the dispersal capacities for one species (please also see a well-known review on amphibian dispersal: Smith, M. A., and D. M. Green. 2005. Dispersal and the metapopulation paradigm in amphibian ecology and conservation: are all amphibian populations metapopulations? *Ecography* 28:110-128⁹). Besides, a large number of species dispersal data are lacking. So, this is not practicable to incorporate the exact dispersal information for each species or a single value into the model to conduct future predictions. Though previous literatures often use two scenarios^{10,11} (i.e., full dispersal and no dispersal) or full dispersal^{3,4,6,12}, we kept only the no-dispersal scenario (more biologically relevant) in this study (**Line 461-465**) as dispersal distances are very low for most amphibian and reptile species, as the referee noted.

Minor comments:

1. Why making such a big deal about the addition of Chinese species? There shouldn't be a political agenda in scientific research, and I would recommend the authors to do the same for other regions where the GBIF dataset is under-represented if they want to maintain their claims.

Response: Thank you for this constructive suggestion. Following the reviewer's concern, we have tried to contact and collaborate with over 30 herpetologists to supplement occurrence records across the world (over 2.5 million records), especially in the under-represented region of online database, such as Central Asia and East Europe (e.g., Uzbekistan, Kazakhstan, Kyrgyzstan, Turkmenistan, Russia, Ukraine, Latvia, Hungary), South Asia (Pakistan, India), Middle East (Turkey, Iran, the Arabian Peninsula.), Africa (Chad, N and S. Sudan, the republics of the Congo), and South America (Bolivia, Paraguay, Argentina, Peru, Colombia), **Line 367-373**. We have re-conducted all our model constructions and data analyses using the new database, and updated all our results throughout the whole text.

2. Authors state that “For records with only location information (village or town), we georeferenced the longitude and latitude using Google Earth v7.1”. But, if you use a 1 x1 km² scale, then this does not work to build models. Your species will be expected to thrive in urban habitat (and the correction for that later in the text only includes “large cities”. We cannot judge on the quality of the data.

Response: Thank you for this very helpful question. For the presence data with village and town information, the exact location name in the literatures are based on the authors’ fieldwork. We therefore georeferenced these location names using Google Earth and identified the sampling sites according to those descriptions. These points are usually far away from urban areas. As requested by reviewers, we should not make the addition of Chinese species, we do not strengthen how we collect Chinese species records no more in this revision, because we have expanded our dataset largely to other countries.

3. You cannot compare families by the number of species in PA, take the Scincidae for instance, that’s about 4000 species (more or less), how can it be compared to family that have a few dozen (or even hundreds) of species. Please use ratios instead.

Response: Thanks for this constructive suggestion. We have revised our manuscript accordingly by using ratios data instead of the number of species data, see **Line 207-214** and **Table S6**.

4. Some minor mistakes in English, related to typos or missing words more than structure.

Response: Thanks and we have checked typos carefully throughout the whole text.

Comments from files attached

1. incorporating the deficient data in China. Why creating a bias in the data? Deficient data are everywhere in the world.

Response: Thank you for this suggestion. In this round of revision, we have cooperated with more than 30 experts in the field of amphibian and reptile research worldwide, and have added new occurrence data over 2.5 million records, especially in the under-represented region such as Central Asia to East Europe, South Asia, Middle East, Africa, South America, etc. Totally we have increased another 1048 amphibian species and 4136 reptile species in our new data analyses.

2. Finer than what? There may be something missing here.

Response: Sorry for the unclear description. Here, we meant that the resolution of 1 km × 1 km we used in the present study is finer than the 10 km × 10 km resolution that most previous global studies used. Because of 150-word limitation in **Abstract** section, we delete this sentence.

3. “over 98% of herpetofauna are distributed in PAs.” Is that the percentage of species? The percentage of biomass?

Response: Thank you for pointing this out. We changed it to “...herpetofauna species ...”, **Line 80**.

4. “Distributional ranges will also undergo less habitat loss inside PAs than outside”, This follows expectations, but under a habitat loss scenario, is it still linked to climate change here?

Response: Yes, it is linked to climate change. We have re-edited the sentence to “Species’ distributional ranges will also undergo less habitat loss inside PAs than outside them”, **Line 81-82**.

5. “with more than 500 terrestrial animals on the brink of extinction”, Quite a few are gone already, that may also be important to highlight.

Response: Thanks. We rewrite this sentence to make it more clear, “...with thousands of species on the brink of extinction and >500 species declared or believed extinct in the last 500 years only among terrestrial vertebrates¹³⁻¹⁶.”, **Line 94-96**.

6. “considered” Does that mean they are considered as such, but they are not for real?

Response: Sorry for the potential confusion. We have deleted “considered as” to make the description more accurate.

7. “The global open-access database of species occurrence records and environmental layers allows for robust prediction of species ranges” The reference here is only an example, not an evidence. It would be better to cite something related to the value of the dataset you used, either for your focal species, or at the global scale – or both. Such publications exist for sure for amphibians.

Response: Thanks. We have replaced the reference using terrestrial vertebrates on a global scale following your suggestion in our revised manuscript. Please see references Liu. et al 2020¹⁷ (Animal invaders threaten protected areas worldwide), Carlson et al 2022⁷ (Climate change increases cross-species viral transmission risk), **Line 119-122.**

8. “Linking species range (habitat) dynamics...”. These two concepts are entirely different! Some of the habitat within a species’ range is not adequate for the survival of the species.

Response: Thanks for pointing this out. We have changed it to “As expected, using species range dynamics to evaluate the role of PAs under climate change”, **Line 122-123.**

9. “in China, the USA, Mexico, South Africa, and Western Europe” Five regions but three references. I recommend linking these to clarify.

Response: Thanks, we have cited it separately to make it more clear as “SDM studies in China³, the USA¹⁰, Mexico¹⁸ and South Africa¹⁸”, **Line 126.**

10. “regarded” So they are not for real?

Response: Thanks. We have deleted “regarded as” to not confuse readers. **Line 128.**

11. “Amphibians and reptiles (hereafter herpetofauna) are regarded as most threatened terrestrial vertebrate taxa under climate change.” The reference used here is old and not relevant. For amphibians, there are many papers that can be cited, for reptiles, please refer to the main publication from last week (I understand the author could not have cited it when originally submitting)

Response: Thank you for your suggestion on the latest reference, which we have cited in our revised manuscript (IUCN 2021¹⁹, Cox et al 2022²⁰), **Line 129.**

12. “with nearly 41% of amphibians and 21% of reptiles” And they still are!, A tiny bit more than that now.

Response: Thanks. These two figures were based on the latest report from the IUCN website (accessed on Aug 14, 2022), showing that the number is still 41% for amphibians and 21% for reptiles.

13. “being classified” change to “listed”

Response: Thanks, we have revised it accordingly, **Line 131.**

14. “1,000,000 observed records” Here it would be good to know if they are geographically independent, and taking only the US and China as example is a bad idea, the world is not polarized around these two regions. What about adding other options in the comparison?

Response: Thanks for this important suggestion. We have added another two countries including Australia (developed country) and Brazil (developing country) as additional representative examples, **Line 141-144**.

15. “This low resolution may not be comparable with current PA resolution (median = 0.59 km²), and overestimate range size of narrow-range species.” Where is this value from, and how do you define a PA in this case? That seems to be a very low value and I wonder if it refers to PAs strictly speaking, or to areas under some sort of conservation policy.

Response: Thanks for the comments. We have updated the median value of the PA area based on the latest information from the World Database of Protected Areas (WDPA) datasets (July 2022; for more details see Methods section, **Line 426-435**). As you will see, the updated median value of the area for terrestrial strict PAs is 0.37 km². Please also see our response to your 4# major comment above.

16. “comprising 4,355 amphibian species and 4,857 reptile species from public databases and published references” This is half of the species in both cases! And the most abundant species are the one for which more datapoints are available, it totally biases the data, and you cannot speak about global herpetofauna in this case, this study is about the abundant herpetofauna.

Response: Thank you for this very constructive comment. In order to address the reviewer’s concern on the number of study species involved in our present study, we have expanded our dataset by collecting additional 2.5 million occurrence records (especially in data-deficient regions), including 1048 amphibian and 4136 reptile species with the aid of over 30 herpetologists worldwide. The updated dataset covers 63.6% (5403/8489; Aug, 2022) of amphibians and 76.6% (8993/11733; Mar, 2022) of reptile species, and includes both abundant species and rare species, with species range sizes from 1 km² to several million km². Importantly, our main conclusions did not change after we expand our dataset, which demonstrates that our results are robust to data uncertainties. Please also see our response to your 1# major comment above.

17. “specifically including data for species distributed in China (Table S2)”, Why including a bias here?

Response: Thanks for this very good input. As you will see, during our revision, we have cooperated with more than 30 experts in the amphibian and reptile field worldwide, and have added occurrence records across the world (over 2.5 million records), especially in those previously under-represented regions such as Central Asia and East Europe (e.g., Uzbekistan, Kazakhstan, Kyrgyzstan, Turkmenistan, Russia, Ukraine, Latvia, Hungary), South Asia (Pakistan, India), Middle East (Turkey, Iran, the Arabian Peninsula.), Africa (Chad, N and S. Sudan, the republics of the Congo), and South America (Bolivia, Paraguay, Argentina, Peru, Colombia), **Line 367-373**.

18. “under the assumption that future land use remains unchanged for this study”. This is quite an important assumption, I recommend you include it in the caption of figures, it could be pretty misleading otherwise.

Response: Thanks for the suggestion. We have clarified this issue in the caption of **Fig. 1-3, and Supplementary Figures**.

19. “evaluate the conservation effectiveness of existing PAs in protecting herpetofauna under current and future climate scenarios.” For abundant species

Response: As we have explained to the reviewer above, the current dataset we used has included more species after our close collaborations with more than 30 scientists in the field of herpetology. Now, we are confident that our species not only included abundant species, but also those rare species with small range size. We have also provided the range size information in the supporting material (**Table S3 and S4**) to clarify this point more clearly.

20. “Next, we conducted an intensive search of occurrence records from China, to update information on data-deficient species” Why China only? What about other countries with high species diversity but low representation in the publicly accessible databases?

Response: Thank you again for this very important suggestion. Please also see our responses to your similar concerns above, and we have largely expanded our datasets, especially in those previously low sampling regions in our new analyses.

21. “For records with only location information (village or town), we georeferenced the longitude and latitude using Google Earth v7.1.” If you use a 1 x1 km km2 scale, then this does not work to build models. Your species will be expected to thrive in urban habitat.

Response: Thank you again for this very helpful suggestion. Please also see our response to your 2# minor comment above on the reason why the points we used are far from urban areas.

22. “We removed problematic records from the GBIF that fell outside the spatial maps offered by the International Union for Conservation of Nature and Natural Resources (IUCN) accessed in March 2019.” This is wrong! The IUCN range maps are not as accurate as other records, especially in China. These datapoints may be problematic for the authors, but not for science! Removing them is not wrong, removing them for this reason is the problem.

Response: Thank you for this constructive comment. In order to account for the potential impact of the IUCN maps on our results, we followed a previous study (Ficetola et al. 2014)²¹ by creating a 400 km buffer zone around the IUCN maps to construct SDMs.

23. “In total, we used five bioclimatic variables to construct our species distribution models: annual mean temperature (BIO1), maximum temperature of the warmest month (BIO5), minimum temperature of the coldest month (BIO6), mean annual precipitation (BIO12), and precipitation during the warmest quarter (BIO18).” Isothermality is critical for many ectotherms. I don’t understand why it was not selected here.

Response: Thank you again for this suggestion. Please also see our response to your 2# major comment. We have re-run SDMs by including the Isothermality as an environmental layer in our new round of data analysis, and we obtained similar results.

24. “We used Representative Concentration Pathways (RCPs) 2.6 and 8.5 as future climate conditions during 2060–2080 (2070)” This is not adequate, please see my review.

Response: Thanks for the suggestion. Please also see our response to your 3# major comment. In our new round analysis, we re-conducted our model analyses under different carbon emission scenarios including RCP 2.6, 4.5, 6.0 and 8.5.

25. “Generalized Linear Model, Generalized Boosted Regression Models, Maximum Entropy, Random Forest, and Support Vector Machines, all commonly used in SDM studies” True, they are commonly used, but they also are selected for good reasons.

Response: Thanks for the suggestion. We change our word to “In sum, we used five commonly used and with high model performance SDM algorithms in the ensemble models: Generalized Linear Model²², Generalized Boosted Regression Models²³, Maximum Entropy²⁴, Random Forest²⁵ and Support Vector Machines²⁶”, **Line 416-418**.

26. “TSS ≥ 0.6 ” This a not a high value, especially for such a sample size. Were the results of the models unsatisfying?

Response: Thank you for this suggestion. In response to your concern, we have used TSS ≥ 0.7 (**Line 421**) in our new round of data analyses, which is a widely used threshold to evaluate model performance (e.g., Wang, B. et al 2017; Gallardo, B. et al. 2018^{27,28}).

27. “with PA maps”, There is a problem here, not all areas from the database are PA strictly speaking. Some are area with regulations, but they are not PAs.

Response: Thanks. Please see our response to your 4# major comments on this issue.

28. we treat the 15% coverage as a summary benchmark of conservation status, Based on which criteria?

Response: Sorry for the misunderstanding. We took 15% as a threshold referring to Zhu et al 2021¹⁰ and Jennings et al. 2020²⁹. Besides, we also changed this target to 30% as a sensitivity test. We have added the reference information in our revised text.

29. “threatened species (i.e., classified as Near Threatened, Vulnerable, Endangered, Critically Endangered, and Extinct based on IUCN; 1,052 and 526 species for amphibians and reptiles, respectively” If re-running the analyses, I recommend updating the reptile dataset following the update last week.

Response: Thanks for the suggestion. We have revised our manuscript according to the latest update, **Line 450-452**.

30. and species have unlimited dispersal capacity (hereafter unlimited-dispersal). How biologically meaningful is that?

Response: Thank you for your input on this important issue. Please also see our response to your 5# major comment above. Considering the fact that most amphibian and reptile species have relatively low dispersal abilities, which also varied among geographical populations, we only used the no-dispersal scenario in our data analyses.

Comments from reviewers 3:

The manuscript focused on an important topic about evaluating the effectiveness of protected areas (PAs) for amphibians and reptiles on a global scale. In general, the study is interesting as it involved nice efforts to collect species records from multiple resources/studies and followed a sophisticated workflow to quantify the effectiveness of PAs and identify gaps in the baseline climate condition and under future climate scenarios. The results, however, seem surprising to me as I found big inconsistencies between them and the results of a similar study (but at the regional level) that we conducted in our lab (not published yet). The authors need to explore and discuss the consistencies/inconsistencies of their results and the other studies (which are missing from the manuscript).

Response: We thank the reviewer very much for providing positive comments on the value of our present study. Regarding your very important suggestion on the comparison of our results with other studies, we have added related sentences in the second and third paragraphs (**Line 266-284**) of the **Discussion** section. In addition to amphibians and reptiles the present study focused on here, our finding has also been observed across other taxonomic groups such as invertebrates and endotherms^{30,31}, despite some other studies have found opposite patterns at continental or regional scales. For example, it has been predicted that current PAs may become less effective under climate change for conserving amphibian biodiversity in Italy³², This discrepancy is likely because the protection offered by PAs for species varies among taxa and regions³³. In addition, this discrepancy might also be caused by different conservation targets and RCP scenarios. However, overall, most regional studies are consistent with our results showing that PAs will be critically important to the challenge of climate change global.

1. One methodological issue that may affect the results is that the SDMs are strongly sensitive to extent of the study area where background (pseudo-absences) records are drawn. To address the issue, the extent can be limited to the areas that are accessible for species to draw background points for each species separately. It means that for each species, the SDMs would be trained using records within the accessible areas. One way to approximate those areas for each species is to use Wallace zoogeographic regions and consider areas where species occurrence records (presence points) are located. The trained SDMs over a limited extent can then be used to predict/project the potential distribution on a global scale as far as the areas have a climate condition within the range used to train the model (i.e., the areas with a condition

outside of the range should be excluded from the predicted map as the SDMs do not perform well for extrapolations).

Response: Thank you very much for this very helpful suggestion. Following your suggestion, we have used the Wallace's Zoogeographic Regions of each species located as the accessible areas as the background area in our new round of SDM construction (**Line 411-413**). We obtained similar trends and an even more important role of PAs on herpetofauna conservation under future climate change.

Some minor issues:

1. Lines 126-127: What do you mean by “the statistical trends of RCP 2.6 and 8.5 were similar”? Do you mean that the effectiveness of PAs would not be affected under the worst future climate scenario compared to “very stringent” pathway (RCP 2.6)?

Response: Sorry for this misunderstanding. Our original description of “the statistical trends of RCP 2.6 and 8.5 were similar” tends to show readers that there are similar patterns of species potential distributions in PAs between the two scenarios. In our revised text, we have further clarified this sentence by removing this sentence. Furthermore, we have provided more details on the comparison of our predicted results under different carbon emission scenarios in **the Methods, Results and Discussion** sections, see **Line 174, 177, 192-193, 237, 276, 279, 403, 464-466**. For instance, the rarity weighted richness, the proportion of suitable habitat in PAs and the percentage of effectively protected species will both be predicted to increase under all four RCPs.

2. Line 133: the “eRandom” method is not activated in the sdm package yet, so it means that you used the “gRandom” method (default).

Response: We thank the reviewer to point this out. We have changed “eRandom” to “gRandom” in our revised method, see **Line 411**.

3. Line 144: modify the sentence “... the threshold by maximum TSS” to “...the threshold that maximizes TSS”.

Response: We have revised it accordingly, see **Line 423**.

4. It would be nice if the codes used to run the workflow are shared in the supplementary, which could be used for a more precise evaluation of the workflow.

Response: Thanks. We have uploaded all the codes used in our data analysis in **Supplementary Files**.

References used in our responses to the reviewers

1. Aiello-Lammens, M. E., Boria, R. A., Radosavljevic, A., Vilela, B. & Anderson, R. P. spThin: an R package for spatial thinning of species occurrence records for use in ecological niche models. *Ecography* 38, 541–545 (2015).
2. Erfanian, M. B., Sagharyan, M., Memariani, F. & Ejtehadi, H. Predicting range shifts of three endangered endemic plants of the Khorassan-Kopet Dagh floristic province under global change. *Sci Rep* 11, 9159 (2021).
3. Chen, Y., Zhang, J., Jiang, J., Nielsen, S. & He, F. Assessing the effectiveness of China's protected areas to conserve current and future amphibian diversity. *Divers Distrib* 23, 146–157 (2017).
4. Brown, J. L., Cameron, A., Yoder, A. D. & Vences, M. A necessarily complex model to explain the biogeography of the amphibians and reptiles of Madagascar. *Nat Commun* 5, 5046 (2014).
5. Li, X., Liu, X., Kraus, F., Tingley, R. & Li, Y. Risk of biological invasions is concentrated in biodiversity hotspots. *Front. Ecol. Environ.* 14, 411–417 (2016).
6. Borzée, A. *et al.* Climate change-based models predict range shifts in the distribution of the only Asian plethodontid salamander: *Karsenia koreana*. *Sci Rep* 9, 11838 (2019).
7. Carlson, C. J. *et al.* Climate change increases cross-species viral transmission risk. *Nature* 607, 555–562 (2022).
8. Nelson, A. & Chomitz, K. M. Effectiveness of Strict vs. Multiple Use Protected Areas in Reducing Tropical Forest Fires: A Global Analysis Using Matching Methods. *Plos One* 6, e22722 (2011).
9. Smith, M. A. & Green, D. M. Dispersal and the metapopulation paradigm in amphibian ecology and conservation: are all amphibian populations metapopulations? *Ecography* 28, 110–128 (2005).
10. Zhu, G., Papeş, M., Giam, X., Cho, S.-H. & Armsworth, P. R. Are protected areas well-sited to support species in the future in a major climate refuge and corridor in the United States? *Biol Conserv* 255, 108982 (2021).
11. Fitzpatrick, M. C. *et al.* Forecasting the future of biodiversity: a test of single- and multi-species models for ants in North America. *Ecography* 34, 836–847 (2011).
12. Franklin, J., Wejnert, K. E., Hathaway, S. A., Rochester, C. J. & Fisher, R. N. Effect of species rarity on the accuracy of species distribution models for reptiles and amphibians in southern California. *Divers Distrib* 15, 167–177 (2009).
13. Ceballos, G. *et al.* Accelerated modern human-induced species losses: Entering the sixth mass extinction. *Sci Adv* 1, e1400253 (2015).

14. Pimm, S. L. *et al.* The biodiversity of species and their rates of extinction, distribution, and protection. *Science* 344, 1246752 (2014).
15. Urban, M. C. Accelerating extinction risk from climate change. *Science* 348, 571–573 (2015).
16. Pincheira-Donoso, D. *et al.* Temporal and spatial patterns of vertebrate extinctions during the Anthropocene. *Biorxiv* 2022.05.05.490605 (2022) doi:10.1101/2022.05.05.490605.
17. Liu, X. *et al.* Animal invaders threaten protected areas worldwide. *Nat Commun* 11, 2892 (2020).
18. Hannah, L. *et al.* Protected area needs in a changing climate. *Front Ecol Environ* 5, 131–138 (2007).
19. IUCN. The IUCN red list of threatened species. <http://www.iucnredlist.org/> (2021).
20. Cox, N. *et al.* A global reptile assessment highlights shared conservation needs of tetrapods. *Nature* 1–6 (2022) doi:10.1038/s41586-022-04664-7.
21. Ficetola, G. F. *et al.* An evaluation of the robustness of global amphibian range maps. *J Biogeogr* 41, 211–221 (2014).
22. Williams, J. N. *et al.* Using species distribution models to predict new occurrences for rare plants. *Divers Distrib* 15, 565–576 (2009).
23. Graham, C. H. *et al.* The influence of spatial errors in species occurrence data used in distribution models. *J Appl Ecol* 45, 239–247 (2008).
24. Elith, J. *et al.* Novel methods improve prediction of species' distributions from occurrence data. *Ecography* 29, 129–151 (2006).
25. Mi, C., Huettmann, F., Guo, Y., Han, X. & Wen, L. Why choose Random Forest to predict rare species distribution with few samples in large undersampled areas? Three Asian crane species models provide supporting evidence. *PeerJ* 5, e2849–e2849 (2017).
26. Drake, J. M., Randin, C. & Guisan, A. Modelling ecological niches with support vector machines. *J Appl Ecol* 43, 424–432 (2006).
27. Wang, B. *et al.* Australian wheat production expected to decrease by the late 21st century. *Glob Change Biol* 24, 2403–2415 (2017).
28. Gallardo, B. *et al.* Protected areas offer refuge from invasive species spreading under climate change. *Glob Change Biol* 23, 5331–5343 (2017).
29. Jennings, M. D. Gap analysis: concepts, methods, and recent results*. *Landscape Ecol* 15, 5–20 (2000).
30. Thomas, C. D. *et al.* Protected areas facilitate species' range expansions. *Proc National Acad Sci* 109, 14063–14068 (2012).

31. Lawson, C. R., Bennie, J. J., Thomas, C. D., Hodgson, J. A. & Wilson, R. J. Active management of protected areas enhances metapopulation expansion under climate change. *Conserv Lett* 7, 111–118 (2014).
32. D'Amen, M. *et al.* Will climate change reduce the efficacy of protected areas for amphibian conservation in Italy? *Biol Conserv* 144, 989–997 (2011).
33. Hole, D. G. *et al.* Projected impacts of climate change on a continent-wide protected area network. *Ecol Lett* 12, 420–431 (2009).

Reviewer comments, second round review –

Reviewer #3 (Remarks to the Author):

The author managed to revise the manuscript and addressed all the comments well. The study is interesting, well-written and well-presented, and is a significant contribution to the field.

I have no further comments, the only minor one is that I think the authors missed uploading the source codes as supplementary as they mentioned they did in their response letter.

Reviewer #4 (Remarks to the Author):

This is an impressive study that evaluates impacts of future climate on the global distributions of amphibians and reptiles in protected areas. The study is the most comprehensive conducted to date for these taxonomic groups. The results will have broad interests to environmentalists and biologists, and likely will also have political and economic ramifications concerning policies to reduce emissions and climate change.

The conclusions of this work are surprisingly reassuring: almost all species are already in protected areas (>91%), and all future warming climate scenarios (including the most extreme) will not alter this proportion at all. However, I do have some concerns about the manuscript in its revised version, which are described below, in order of when they first appear in the text.

- 1) Lines 77-79. Provide a brief caveat in the abstract, that rare and small ranged species (36.4% amphibians, 23.6% of reptiles) were excluded from this study (see lines 356-359). This is crucially important, as this excluded group are likely the most vulnerable to climate change.
- 2) Line 151. Is it really the case that the median area of all protected areas is just 0.37 km²? This is a tiny area- just 37 ha. This does not agree with my experience working with protected areas across three continents. And for many species, a 37 ha protected area is probably not sufficient for long term survival. If this number is real, then tiny protected areas such as municipal parks have probably been included, which will have almost no impact on conservation. In this case, a minimum threshold of area is needed to only include protected areas that are of meaningful size.
- 3) Lines 182-202 and Fig 2 A, B. Species range shifts with climate change always result in some species increasing their range size- the species 'winners' of climate change. Yet these are not mentioned at all here in this text. Fig 2A shows that 75% of amphibians and reptiles had at least a 23% range loss inside or outside protected areas. Scanning through Table S3 and S4 I could not find a single species that has expanded its range due to climate change. This seems to be an unusually high number of species responding negatively to climate change, and especially so for reptiles. The authors should compare their results to other studies, and discuss what is happening with their results.
- 4) Lines 211-212. The statement "the monotypic snake family Xenotyphlopidae has the greatest percentage of species not represented in PAs currently (100%)" has the following two problems, that may indicate errors in the analyses. Firstly, although the authors state all species has ≥ 5 localities for distribution modelling, and give their sources (lines 366-367), I could not find any localities for the species *Xenotyphlops* (or former genus *Typhlops*) *grandidieri* (or junior synonym *mocquardi*) at these sources (this is the only species in the family *Xenotyphlopidae*). Using the primary literature I found four localities. So what are the ≥ 5 localities and what is the source? I cannot determine this, as the locality data for each species are not provided by the authors in any of the supplementary files or other public repository. The second issue is that one of the four published localities is in a protected area: Oronjia National Park. And this park is included in the protected area data set (WDPA) used by the authors (line 426). So how did the species distribution models not include this species (and monotypic snake family) in this protected area?

5) Line 289. Give a citation or data source to support this statement.

6) Lines 356-359. This important statement should also include the data coverage given in lines 383-384: 'Our data set covers 63.6% of amphibians and 76.6% of reptiles'. The paper would be further strengthened by summarizing the IUCN extinction risks for these excluded rare and small ranged species. These are the species most vulnerable to range shifts from climate change, so they should not be ignored in this global study.

7) Lines 365-367. It is well known that these databases include problems of misidentified taxa, and errors in localities (that have mostly not been proofed). Removing records > 400 km from IUCN maps (line 373) will partly remove geographic errors. But for sure this data set will still include a lot of error, which will result in species distribution models being larger than actual species distributions, thus erroneously placing species in protected areas that they do not occupy. This bias in the results needs to be addressed in this study.

8) Lines 363-379. Because all this work is based on uniquely assembled species distribution data, it is essential that these locality data are made available to the readers of this study.

9) Line 408. This work produced 71,980 species distribution models (14,396 species x 5 climate scenarios), yet none are shown in the supplementary data. It would be helpful to see example models for different taxa, at different scales, and for different regions/biomes.

10) Line 469-471. This statement is inaccurate- there is almost no raw data available in the article or the supplementary files. The raw locality data must be made available as supplementary files to this paper, or deposited in an open access depository such as FigShare.

11) No mention is made anywhere of any of the 14,396 species going extinct under the future climate change scenarios. However, Table S3 and S4 gives the species total distribution areas for each future scenario, and quite a few species go to zero total area for all future climate scenarios. And yet more species have ranges reduced to < 10km². These results do not surprise me. However, I am surprised that this is never discussed in this paper. The prediction of this degree of species extinction from climate change is an incredibly important finding from this study, and strongly runs counter to the rosy picture painted by the abstract.

Reviewer #5 (Remarks to the Author):

The manuscript, "Global Protected Areas as refuges for amphibians and reptiles under climate change" by Mi et al., is a good review of herpetofauna in protected areas that will be affected by climate change. Although the results are not surprising and there are few positive proactive suggestions to maximize conservation of herpetofaunal species overall, the authors have done a solid job responding to reviewers. I was asked to specifically look into reviewer #2's comments and the responses by the authors and I have focused on this activity for this review. Almost all of the comments are well received and the authors were able to revised in a reasonable manner with adequate changes to the initial manuscript. Good job revising the manuscript in light of the suggestions from your reviewers. I find only good things to write about the authors' responses to reviewers' comments. The revised manuscript, however, still suffers from too many run-on sentences, long-winded cadence of sentence structure in many places, and some English grammar inconsistencies. Overall, the work is scientifically sound at this stage and I would suggest only some superficial editorial work to improve word choice and language flow to increase clarity, perhaps with an eye for concise sentence structure. Good job overall.

**Global Protected Areas as refuges for amphibians and reptiles under climate change
(NCOMMS-22-13511B)**

Comments from reviewer 3:

The author managed to revise the manuscript and addressed all the comments well. The study is interesting, well-written and well-presented, and is a significant contribution to the field. I have no further comments, the only minor one is that I think the authors missed uploading the source codes as supplementary as they mentioned they did in their response letter.

Response: Thank you very much for your positive and constructive comments. We have uploaded the source codes in 10.6084/m9.figshare.20958190.

Comments from reviewer 4:

This is an impressive study that evaluates impacts of future climate on the global distributions of amphibians and reptiles in protected areas. The study is the most comprehensive conducted to date for these taxonomic groups. The results will have broad interests to environmentalists and biologists, and likely will also have political and economic ramifications concerning policies to reduce emissions and climate change. The conclusions of this work are surprisingly reassuring: almost all species are already in protected areas (>91%), and all future warming climate scenarios (including the most extreme) will not alter this proportion at all. However, I do have some concerns about the manuscript in its revised version, which are described below, in order of when they first appear in the text.

Response: We appreciate very much the reviewer's positive and constructive comments, and finding that our results may be helpful with the future conservation of amphibian and reptiles in protected areas. We have addressed your concerns and comments below.

1) Lines 77-79. Provide a brief caveat in the abstract, that rare and small ranged species (36.4% amphibians, 23.6% of reptiles) were excluded from this study (see lines 356-359). This is crucially important, as this excluded group are likely the most vulnerable to climate change.

Response: We have added this information in the Abstract now, "we collated distributional data for >14,000 (~70% of) species of amphibians and reptiles" (Lines 78).

2) Line 151. Is it really the case that the median area of all protected areas is just 0.37 km²? This is a tiny area- just 37 ha. This does not agree with my experience working with protected areas across three continents. And for many species, a 37 ha protected area is probably not sufficient for long term survival. If this number is real, then tiny protected areas such as municipal parks have probably been included, which will have almost no impact on conservation. In this case, a minimum threshold of area is needed to only include protected areas that are of meaningful size.

Response: Thank you very much for your careful review on the size of the protected areas we used in the study. To further address your concern, we have recalculated the median value of global PAs' areas from the World Database of Protected Areas (WDPA) dataset (Class I to IV), and got the same number. The median area of PAs is so small because lots of very small PAs exist in Europe, North America, and Australia. For example, Bushy Island in Australia, Ulm Peak in the USA, and Storkollen in Norway, these PAs have a very small areas but are

identified as Class I according to the IUCN category. In our analysis, no matter we used the strict PAs (Class I to IV) or all PAs (Class I to VI), these small-range PAs were included. We undertook sensitive analyses, which excluded PAs whose areas $< 1 \text{ km}^2$ or 5 km^2 , and found our results did not change due to the inclusion of these small-range PAs (Fig. 1 and 2): most species occur in PAs, the Rarity-weighted Richness in PAs will increase in the future (Fig. 1); species' ranges outside PAs will lose more habitat than inside PAs (Fig. 2); meanwhile, the proportion of species with over 15% of their range covered by current PA networks will increase for both amphibians and reptiles (Fig. 2).

Fig.1 Percent of species that have suitable habitats in protected areas (PAs) after excluding PAs' with areas $< 1 \text{ km}^2$ (first row), and $< 5 \text{ km}^2$ (second row).

Fig. 2 Climate change impacts on the percentage of species range (area of habitat) inside and outside PAs by 2070 (RCP 4.5) after excluding PAs' with areas < 1 km² (first row), and < 5 km² (second row).

3) Lines 182-202 and Fig 2A, B. Species range shifts with climate change always result in some species increasing their range size- the species ‘winners’ of climate change. Yet these are not mentioned at all here in this text. Fig 2A shows that 75% of amphibians and reptiles had at least a 23% range loss inside or outside protected areas. Scanning through Table S3 and S4 I could not find a single species that has expanded its range due to climate change. This seems to be an unusually high number of species responding negatively to climate change, and especially so for reptiles. The authors should compare their results to other studies, and discuss what is happening with their results.

Response: Thanks for the comments. We are sorry that we did not clarify this important issue that may have confused the reviewer. In this study, we modeled all species' suitable area dynamics with no dispersal capacity, which means species ranges that will not expand under climate change in this study. Actually, in the first version of MS, we included models with “species with unlimited dispersal capacity”. However, the Reviewer 2 questioned the biological relevance of models with unlimited dispersal capacity, given the low dispersal capacity of amphibians and reptiles. Following the suggestion from Reviewer 2, we removed the analysis with unlimited dispersal capacity. Instead, we explained why we used models excluding species dispersal capacity in **line 162-164** “Because of the limited dispersal ability

of amphibians and reptiles, we assumed no occupation of newly emerged suitable habitat conditions that may become available (e.g., due to climate change) in the future” and in **Lines 496-500 we write** “Most amphibian and reptile species, especially salamanders, have weak dispersal abilities^{50,51}, the dispersal distance differs among populations⁵⁰, which are difficult to be controlled in SDM construction. We therefore calculated all metrics assuming no dispersal capacity under four RCP scenarios (RCPs 2.6, 4.5, 6.0, and 8.5). This means that, under our assumptions, no range expansions can occur.”

Following your suggestion, we add sentences to the Discussion to acknowledge that “Noteworthy, we projected species distribution without considering the dispersal capacity of species, and this projection may underestimate range expansions of species under climate change.” in **Lines 278-280**. And we also compare our results with other studies see **Line 280-299**.

4) Lines 211-212. The statement “the monotypic snake family Xenotyphlopidae has the greatest percentage of species not represented in PAs currently (100%)” has the following two problems, that may indicate errors in the analyses. Firstly, although the authors state all species has ≥ 5 localities for distribution modelling, and give their sources (lines 366-367), I could not find any localities for the species *Xenotyphlops* (or former genus *Typhlops*) *grandidieri* (or junior synonym *mocquardi*) at these sources (this is the only species in the family *Xenotyphlopidae*). Using the primary literature I found four localities. So what are the ≥ 5 localities and what is the source? I cannot determine this, as the locality data for each species are not provided by the authors in any of the supplementary files or other public repository. The second issue is that one of the four published localities is in a protected area: Oronjia National Park. And this park is included in the protected area data set (WDPA) used by the authors (line 426). So how did the species distribution models not include this species (and monotypic snake family) in this protected area?

Response: We are sorry for making you confused. To make the data source clearer, we have now uploaded a list of literature sources for occurrence records as a supplementary file (Supplementary Table 1).

As for the first question, we re-checked the records for *Xenotyphlops grandidieri* and found that we did collate 5 localities for this species (see **Table 1**) rather than 4 localities found by the reviewer. Three records are from Caetano et al. 2022¹, another two different points are from Brown et al. 2014².

As for the second question, in the first version of MS, the Reviewer 2 questioned the areas we used from WDPA. She/he thought not all areas from the database are PA strictly. Some are area with regulations, but they are not PAs. She/he suggested that we should use strict PAs. Therefore, in the revision, we only included strict PAs (IUCN category I to IV) and take PAs (category I to VI) as a sensitive test. We excluded the Oronjia National Park in our analysis as its PA category has not been reported in the WDPA database. Therefore, our analysis did not include *X. grandidieri* in a protected area, although this species is distributed in a national park as found by the reviewer.

Orono	D	Special Areas of Conservat	Regional	Not Reported	Not Applicable	0	0	0.057833
Orono	Stewardship Area	Stewardship Area	National	III	Not Applicable	2	0	0.26575
Oronjia	National Park	National Park	National	Not Reported	Not Applicable	0	0	0.232012
Oronwilm Luomonsuojelualue	Tehtyminen Luomonsuojelu	Private Nature Reserve	National	IV	Not Applicable	0	0	0
Orono Land Trust	Private Conservation	Private Conservation	National	V	Not Applicable	0	0	0
Orono Land Trust 1	Private Conservation	Private Conservation	National	V	Not Applicable	0	0	0
Oronoco Prairie State Scientific and	Natural Area	Natural Area	National	V	Not Applicable	0	0	0
Oronovay	Nature Reserve	Nature Reserve	National	IV	Not Applicable	0	0	0.432993
Oronovay And South Colossay	Site Of Special Scientific	Site Of Special Scientific	National	IV	Not Applicable	1	0	5.122297
OROPEDIO POLOIS	C	Special Protection Area (B	Regional	Not Reported	Not Applicable	0	0	0
OROPEDIO POLOIS	C	Special Areas of Conservat	Regional	Not Reported	Not Applicable	0	0	0
Orup Forest	Stewardship Area	Stewardship Area	National	III	Not Applicable	0	0	0

In order to provide the reviewer a full picture of the species distribution and PA locations, we have provided the exact information below for your review. We found the nearest strict Protected Areas (PAs, IUCN category I to IV) is Makira (category II), which is about 206 km away from the nearest occurrence records (**Table 1, Column 3**); we also found no potential suitable area for *X. grandidieri* is in strict PAs. If all PAs (IUCN category I to VI) are used, the nearest PA is Ambodivahibe (Class V), which is 0.56 km from the nearest occurrent points (**Table 1, Column 4**), and there are 103 km² suitable areas in PAs. Our finding that this species does not occur in any PAs is based on our use of strict PAs, that's why we found that all habitats of *X. grandidieri* were outside the boundaries of PAs.

Table 1 Coordinates of *Xenotyphlops grandidieri* and the spatial location with PAs.

Longitude	Latitude		
49.43889	-12.39028		
49.3925	-12.27333		
49.2896	-12.35744		
49.83817	-12.80658		
49.40703	-12.47487		

5) Line 289. Give a citation or data source to support this statement.

Response: Thanks, we have cited Supplementary Fig. 20, “precipitation in PAs is higher than outside and is predicted to increase in the future (Supplementary Fig. 20)”, **Line 304-305**.

6) Lines 356-359. This important statement should also include the data coverage given in lines 383-384: ‘Our data set covers 63.6% of amphibians and 76.6% of reptiles’. The paper would be further strengthened by summarizing the IUCN extinction risks for these excluded rare and small ranged species. These are the species most vulnerable to range shifts from climate change, so they should not be ignored in this global study.

Response: Thanks for the valuable suggestion. We have added a description at the end of the paragraph “Occurrence records”, “However, many rare, small-ranged, species were not included in this study as they do not currently have enough occurrence records available. These species are already more threatened than included species (see above), are probably less well covered by existing PAs, and may be more vulnerable to range shifts from climate change⁸¹. Our results are thus mostly applicable for the 64%-77% of species with overall larger ranges.”, **Line 413-418**.

Besides, we analyzed the IUCN threatened status of species that were not include in the study, and found 70.5% of amphibians and 64.7% of reptiles are either threatened or data deficient. We report this important information in the Discussion as the reviewer suggested, writing “Finally, it is necessary to highlight that one potential caveat with the present study is that many rare, small-ranged, species were excluded from our analyses as they currently lack sufficient distributional records to construct SDMs. Most of these unanalyzed species, 70.5% of amphibians and 64.7% of reptiles, are assessed as threatened or data deficient according to

IUCN (compared to only 22.9% of amphibians and 16.3% of reptiles included in our dataset). Consequently, further attention to the plight of these species is thus needed when the importance of PAs for their conservation is assessed in the future, because these species are more likely to be at a high extinction risk.”, **Line 381-388.**

7) Lines 365-367. It is well known that these databases include problems of misidentified taxa, and errors in localities (that have mostly not been proofed). Removing records > 400 km from IUCN maps (line 373) will partly remove geographic errors. But for sure this data set will still include a lot of error, which will result in species distribution models being larger than actual species distributions, thus erroneously placing species in protected areas that they do not occupy. This bias in the results needs to be addressed in this study.

Response: To solve the problems of misidentified taxa, we used taxonomic harmonization and normalization on the basis of the GBIF taxonomic backbone to harmonize all species names. To correct potential errors in localities, we removed records from the occurrence records that fell outside the 400 km buffer of the species polygon maps, following the previous study of Ficetola et al. 2014 as suggested by the Reviewer 2 in the first round of revision. In addition, we used the ‘CoordinateCleaner’ package implemented in R to remove records from capitals, institutes and museums. More importantly, we have invited a global team of herpetologists as co-authors to validate the quality of distribution data on amphibians and reptiles across the world. We have done our best to control the data quality, and believe that the error, if any, has been minimized in our dataset. If the reviewer can recommend better methods to increase the accuracy of the data, please let us know and we will gladly consider any better approaches.

8) Lines 363-379. Because all this work is based on uniquely assembled species distribution data, it is essential that these locality data are made available to the readers of this study.

Response: We have uploaded all online data sources (> 86% of all records) in [10.6084/m9.figshare.20958190](https://doi.org/10.6084/m9.figshare.20958190), and we have uploaded a literature source in supplementary files (Supplementary Table 1). Because this study is an international cooperative work, some data (especially for field data) is owned by different authors or agencies, and we have no rights to publish these data (e.g., records in New Zealand, and Saudi Arabia). But readers can mail the corresponding author (Prof. Weiguo Du), and he will help readers to contact the data owner for a request.

9) Line 408. This work produced 71,980 species distribution models (14,396 species x 5 climate scenarios), yet none are shown in the supplementary data. It would be helpful to see example models for different taxa, at different scales, and for different regions/biomes.

Response: Thanks for the suggestion. We now provide an example that how to make a species distribution model for different taxa and Wallace's Zoogeographic Regions automatically in the supplementary file (**Supplementary Example**), **Line 458-459**.

10) Line 469-471. This statement is inaccurate- there is almost no raw data available in the article or the supplementary files. The raw locality data must be made available as supplementary files to this paper, or deposited in an open access depository such as FigShare.

Response: We have uploaded all online data sources (> 86% of all records) in 10.6084/m9.figshare.20958190, and uploaded a literature source in supplementary files (Supplementary Table 1). Because this study is an international cooperative work, some data (especially for field data) is owned by different authors or agencies, and we have no rights to publish these data (e.g., records in New Zealand, and Saudi Arabia). But readers can mail the corresponding author, and he will help the reader to contact the data owner for a request.

11) No mention is made anywhere of any of the 14,396 species going extinct under the future climate change scenarios. However, Table S3 and S4 gives the species total distribution areas for each future scenario, and quite a few species go to zero total area for all future climate scenarios. And yet more species have ranges reduced to < 10 km². These results do not surprise me. However, I am surprised that this is never discussed in this paper. The prediction of this degree of species extinction from climate change is an incredibly important finding from this study, and strongly runs counter to the rosy picture painted by the abstract.

Response: Thank you for raising this important point. To reflect the full picture provided in the results, we have added a sentence to the Abstract “Despite these findings, over 300 amphibian and 500 reptile species are predicted to go extinct under climate change over the course of the ongoing century.” (Lines 85-87). We further added the following sentence to the Result section: “However, our models predict that 359 to 770 amphibian species and 545 to 1098 reptile species will go extinct under different climate change scenarios over the course of the ongoing century (Supplementary Table 4 and 5).” (Line 188-190). We further added the following sentences to the Discussion “However, over 300 amphibian and 500 reptile species are predicted to go extinct due to climate change over the course of the ongoing century. These

were not counted when we calculate the proportion of species covered in PAs in the future, hence our finding - that a large majority of species will be protected in the future relates to surviving species and should not be taken to mean that climate change will not have devastating effects on many amphibian and reptile species.” (Lines 308-313).

Comments from reviewers 5:

The manuscript, "Global Protected Areas as refuges for amphibians and reptiles under climate change" by Mi et al., is a good review of herpetofauna in protected areas that will be affected by climate change. Although the results are not surprising and there are few positive proactive suggestions to maximize conservation of herpetofaunal species overall, the authors have done a solid job responding to reviewers. I was asked to specifically look into reviewer #2's comments and the responses by the authors and I have focused on this activity for this review. Almost all of the comments are well received and the authors were able to revised in a reasonable manner with adequate changes to the initial manuscript. Good job revising the manuscript in light of the suggestions from your reviewers. I find only good things to write about the authors' responses to reviewers' comments. The revised manuscript, however, still suffers from too many run-on sentences, long-winded cadence of sentence structure in many places, and some English grammar inconsistencies. Overall, the work is scientifically sound at this stage and I would suggest only some superficial editorial work to improve word choice and language flow to increase clarity, perhaps with an eye for concise sentence structure. Good job overall.

Response: We thank the reviewer very much for providing positive comments on the value of our present study. We have further improved the scientific writing with the help of our co-authors with English as their native language.

References used in our responses to the reviewers

1. Caetano, G. H. de O. *et al.* Automated assessment reveals that the extinction risk of reptiles is widely underestimated across space and phylogeny. *Plos Biol* 20, e3001544 (2022).
2. Brown, J. L., Cameron, A., Yoder, A. D. & Vences, M. A necessarily complex model to explain the biogeography of the amphibians and reptiles of Madagascar. *Nat Commun* 5, 5046 (2014).

Reviewer comments, third round review –

Reviewer #4 (Remarks to the Author):

NCOMMS 22 13511B

Reviewer 4

I appreciate the hard work and detailed responses provided by the authors, and commend them for being responsive to the prior critiques. In particular:

- 1) Providing clear caveats in the abstract and main paper on the excluded species with small distributions and low number of localities.
- 2) Including new analyses using larger area reserves only (1+ and 5+ km²).
- 3) Clarifying (to me) that these results allow for no dispersion at all.
- 4) Including text reporting the predicted species extinctions associated with future climate change.

However, I still have concerns about the following issues.

A) There still are problems with the protected area being used in the analysis for *X. grandidieri*. Oronjia National Park is an IUCN category V protected area (Goodman et al. 2018). And there are many other strict protected areas of category 1 to IV that are closer to the *X. grandidieri* localities than Makira. The closest is Analamerana Special Reserve (category IV), which is almost certainly included in the suitable area for this species (being at the midpoint between the known localities). This protected area is also listed in the WDPA data base with its correct IUCN category IV.

B) For the problems of misidentified taxa, applying taxonomic harmonization and normalization will not fix this. The problem is not different taxonomies, but species being identified as the wrong species and so producing errors in the localities. Removing localities outside a 400 km buffer will catch large errors, but 400 km is still a huge distance. One solution would be to compare localities to the IUCN Red List area polygons for each species, and remove all localities outside the polygon. At a minimum, this source of error in the data needs to be acknowledged, because it inflates the area of distribution, and so will create bias in the results.

C) The locality data for the species included in this study must be published as supporting data. For the few cases of sensitive species, these localities can be made less precise, to protect sites. These locality data are as essential to this study as molecular sequence data is to genetic work. No journal will allow you to publish a genetic study without providing access to the data. Without the locality data being made accessible to the scientific community, the work is unrepeatable and cannot be assessed. Providing a list of publications and online sources (which will continue to evolve), and withholding other data is not acceptable. It will be impossible to produce a replicate data source. If the authors have no right to publish some of these locality data, or are reluctant to share data, then this should be removed from the analyses.

**Global Protected Areas as refuges for amphibians and reptiles under climate change
(NCOMMS-22-13511B)**

Comments from reviewer 4:

A) There still are problems with the protected area being used in the analysis for *X. grandidieri*. Oronjia National Park is an IUCN category V protected area (Goodman et al. 2018). And there are many other strict protected areas of category 1 to IV that are closer to the *X. grandidieri* localities than Makira. The closest is Analamerana Special Reserve (category IV), which is almost certainly included in the suitable area for this species (being at the midpoint between the known localities). This protected area is also listed in the WDPA data base with its correct IUCN category IV.

Response: We agreed with reviewers' comments, and added a sentence to describe our work limitation for this point. "In addition, more occurrence records collected for data-deficiency species in future and the update of PAs from the WDPA database may influence species distributes in PAs and therefore optimal conservation plans.", Lines 385-387.

B) For the problems of misidentified taxa, applying taxonomic harmonization and normalization will not fix this. The problem is not different taxonomies, but species being identified as the wrong species and so producing errors in the localities. Removing localities outside a 400 km buffer will catch large errors, but 400 km is still a huge distance. One solution would be to compare localities to the IUCN Red List area polygons for each species, and remove all localities outside the polygon. At a minimum, this source of error in the data needs to be acknowledged, because it inflates the area of distribution, and so will create bias in the results.

Response: Thanks for the suggestion, we modified a sentence to describe our work limitation of data source "Although it might inflate the area of species distribution, we removed records from the occurrence records that fell outside the 400 km buffer of the species polygon maps following Ficetola et al. to correct potential errors of occurrence records in databases" Lines 409-411.

C) The locality data for the species included in this study must be published as supporting data. For the few cases of sensitive species, these localities can be made less precise, to protect sites. These locality data are as essential to this study as molecular sequence data is to genetic work.

No journal will allow you to publish a genetic study without providing access to the data. Without the locality data being made accessible to the scientific community, the work is unrepeatable and cannot be assessed. Providing a list of publications and online sources (which will continue to evolve), and withholding other data is not acceptable. It will be impossible to produce a replicate data source. If the authors have no right to publish some of these locality data, or are reluctant to share data, then this should be removed from the analyses.

Response: Thanks for the comments. We revised our description of data availability:

All online occurrence records are available at

<https://doi.org/10.6084/m9.figshare.20958190.v1>. Some occurrence records are available under restricted access for avoiding potential threat of poaching, access can be obtained by contacting the data owners, who have been listed in our **Supplementary Data 7**.